# Cortical encoding of acoustic and linguistic rhythms in spoken narratives

Cheng Luo[1], Nai Ding[1,2]*

[1]Key Laboratory for Biomedical Engineering of Ministry of Education, College of Biomedical Engineering and Instrument Sciences, Zhejiang University, Hangzhou, China; [2]Research Center for Advanced Artificial Intelligence Theory, Zhejiang Lab, Hangzhou, China

**Abstract** Speech contains rich acoustic and linguistic information. Using highly controlled speech materials, previous studies have demonstrated that cortical activity is synchronous to the rhythms of perceived linguistic units, for example, words and phrases, on top of basic acoustic features, for example, the speech envelope. When listening to natural speech, it remains unclear, however, how cortical activity jointly encodes acoustic and linguistic information. Here we investigate the neural encoding of words using electroencephalography and observe neural activity synchronous to multi-syllabic words when participants naturally listen to narratives. An amplitude modulation (AM) cue for word rhythm enhances the word-level response, but the effect is only observed during passive listening. Furthermore, words and the AM cue are encoded by spatially separable neural responses that are differentially modulated by attention. These results suggest that bottom-up acoustic cues and top-down linguistic knowledge separately contribute to cortical encoding of linguistic units in spoken narratives.

*For correspondence:
ding_nai@zju.edu.cn

## Introduction

When listening to speech, low-frequency cortical activity in the delta (<4 Hz) and theta (4–8 Hz) bands is synchronous to speech (*Keitel et al., 2018*; *Luo and Poeppel, 2007*). However, it remains debated what speech features are encoded in the low-frequency cortical response. A large number of studies have demonstrated that the low-frequency cortical response tracks low-level acoustic features in speech, for example, the speech envelope (*Destoky et al., 2019*; *Ding and Simon, 2012*; *Koskinen and Seppä, 2014*; *Di Liberto et al., 2015*; *Nourski et al., 2009*; *Peelle et al., 2013*). Since the theta-band speech envelope provides an important acoustic cue for syllable boundaries, it has been hypothesized that neural tracking of the theta-band speech envelope is a mechanism to segment continuous speech into discrete units of syllables (*Giraud and Poeppel, 2012*; *Poeppel and Assaneo, 2020*). In other words, the theta-band envelope-tracking response reflects an intermediate neural representation linking auditory representation of acoustic speech features and phonological representation of syllables. Consistent with this hypothesis, it has been found that neural tracking of speech envelope is related to both low-level speech features (*Doelling et al., 2014*) and perception. On one hand, it can occur when speech recognition fails (*Etard and Reichenbach, 2019*; *Howard and Poeppel, 2010*; *Peña and Melloni, 2012*; *Zoefel and VanRullen, 2016*; *Zou et al., 2019*). On the other hand, it is strongly modulated by attention (*Zion Golumbic et al., 2013*; *Kerlin et al., 2010*) and may be a prerequisite for successful speech recognition (*Vanthornhout et al., 2018*).

Speech comprehension, however, requires more than syllabic-level processing. Previous studies suggest that low-frequency cortical activity can also reflect neural processing of higher-level linguistic units, for example, words and phrases (*Buiatti et al., 2009*; *Ding et al., 2016a*; *Keitel et al., 2018*), and the prosodic cues related to these linguistic units, for example, delta-band speech

envelope and pitch contour (*Bourguignon et al., 2013*; *Li and Yang, 2009*; *Steinhauer et al., 1999*). One line of research demonstrates that, when listening to natural speech, cortical responses can encode word onsets (*Brodbeck et al., 2018*) and capture semantic similarity between words (*Broderick et al., 2018*). It remains to be investigated, however, how bottom-up prosodic cues and top-down linguistic knowledge separately contribute to the generation of these word-related responses. Another line of research selectively focuses on top-down processing driven by linguistic knowledge. These studies demonstrate that cortical responses are synchronous to perceived linguistic units, for example, words, phrases, and sentences, even when acoustic correlates of these linguistic units are not available (*Ding et al., 2016a*; *Ding et al., 2018*; *Jin et al., 2018*; *Makov et al., 2017*). Based on these results, it has been hypothesized that low-frequency cortical activity can reflect linguistic-level neural representations that are constructed based on internal linguistic knowledge instead of acoustic cues (*Ding et al., 2016a*; *Ding et al., 2018*; *Meyer et al., 2020*). Nevertheless, to dissociate linguistic units with the related acoustic cues, most of these studies present speech as an isochronous sequence of synthesized syllables, which organizes into a sequence of unrelated words and sentences. Therefore, it remains unclear whether cortical activity can synchronize to linguistic units in natural spoken narratives, and how it is influenced by bottom-up acoustic cues and top-down linguistic knowledge.

Here we first asked whether cortical activity could reflect the rhythm of disyllabic words in semantically coherent stories. The story was either naturally read or synthesized as an isochronous sequence of syllables to remove acoustic cues for word boundaries (*Ding et al., 2016a*). We then asked how the neural response to disyllabic words was influenced by acoustic cues for words. To address this question, we amplitude modulated isochronous speech at the word rate and tested how this word-synchronized acoustic cue modulated the word response. Finally, since previous studies have shown that cortical tracking of speech strongly depended on the listeners' task (*Ding and Simon, 2012*; *Zion Golumbic et al., 2013*; *O'Sullivan et al., 2015*), we designed two tasks during which the neural responses were recorded. One task required attentive listening to speech and answering of comprehension questions afterwards, while in the other task participants were engaged in watching a silent movie while passively listening to speech.

## Results

### Neural encoding of words in isochronously presented narratives

We first presented semantically coherent stories that were synthesized as an isochronous sequence of syllables (*Figure 1A*, left). To produce a metrical structure in stories, every other syllable was designed to be a word onset. More specifically, the odd terms in the metrical syllable sequence always corresponded to the initial syllable of a word, that is, word onset, while the even terms corresponded to either the second syllable of a disyllabic word (73% probability) or a monosyllabic word (23% probability). In the following, the odd terms of the syllable sequence were referred to as $\sigma 1$, and the even terms as $\sigma 2$. Since syllables were presented at a constant rate of 4 Hz, the neural response to syllables was frequency tagged at 4 Hz. Furthermore, since every other syllable in the sequence was the onset of a word, neural activity synchronous to word onsets was expected to show a regular rhythm at half of the syllabic rate, that is, 2 Hz (*Figure 1A*, right).

As a control condition, we also presented stories with a nonmetrical structure (*Figure 1B*, left). These stories were referred to as the nonmetrical stories in the following. In these stories, the word duration was not controlled and $\sigma 1$ was not always a word onset. Given that the word onsets in these stories did not show rhythmicity at 2 Hz, neural activity synchronous to the word onsets was not frequency tagged at 2 Hz (*Figure 1B*, right).

When listening to the stories, one group of participants was asked to attend to the stories and answer comprehension questions presented at the end of each story. The task was referred to as story comprehension task, and the participants correctly answered 96 ± 9% and 94 ± 9% questions for metrical and nonmetrical stories, respectively. Another group of participants, however, were asked to watch a silent movie while passively listening to the same set of stories as those used in the story comprehension task. The silent movie was not related to the auditorily presented stories, and the participants did not have to answer any speech comprehension question. The task was referred to as movie watching task. The electroencephalogram (EEG) responses to isochronously presented

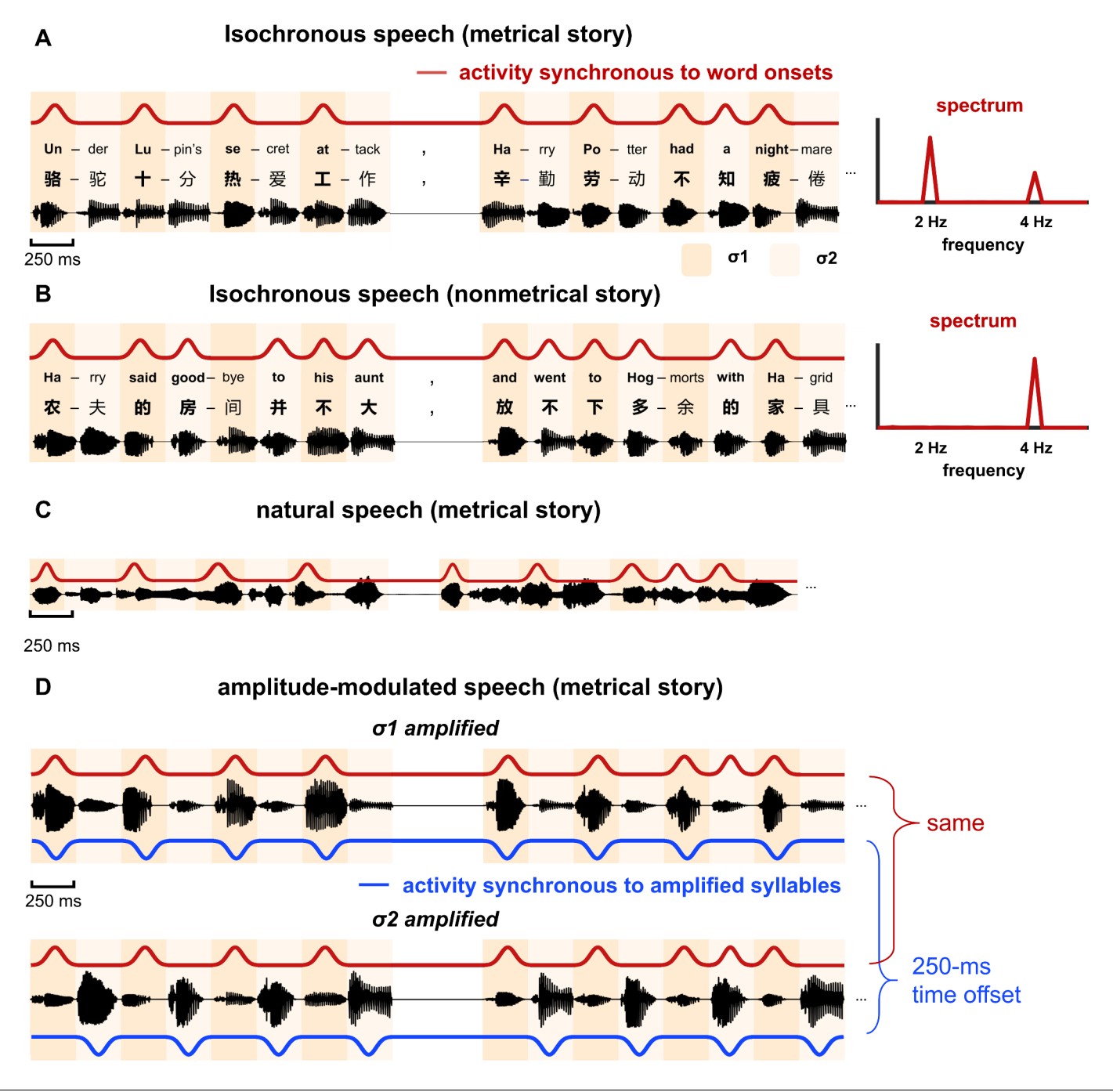

**Figure 1.** Stimulus. (**A and B**) Two types of stories are constructed: metrical stories and nonmetrical stories. (**A**) Metrical stories are composed of disyllabic words and pairs of monosyllabic words, so that the odd terms in the syllable sequence (referred to as σ1) must be the onset of a word. Here the onset syllable of each word is shown in bold. All syllables are presented at a constant rate of 4 Hz. A 500 ms gap is inserted at the position of any punctuation. The red curve illustrates cortical activity synchronous to word onsets, and it shows a 2-Hz rhythm, which can be clearly observed in the spectrum shown on the right. The stories are in Chinese and English examples are shown for illustrative purposes. (**B**) In the nonmetrical stories, word onsets are not regularly positioned, and activity that is synchronous to word onsets does not show 2-Hz rhythmicity. (**C**) Natural speech. The stories are naturally read by a human speaker and the duration of syllables is not controlled. (**D**) Amplitude-modulated isochronous speech is constructed by amplifying either σ1 or σ2 by a factor of 4, creating a 2-Hz amplitude modulation. The red and blue curves illustrate responses that are synchronous to word onsets and amplified syllables, respectively. The response synchronous to word onsets is identical for σ1- and σ2-amplified speech, that is, the phase difference was 0° at 2 Hz. In contrast, the response synchronous to amplified syllables is offset by 250 ms between conditions, that is, the phase difference was 180° at 2 Hz.

stories are shown in *Figure 2A and C*. The response spectrum was averaged over participants and EEG electrodes.

Figure 2A shows the responses from the participants attentively listening to the stories in story comprehension task. For metrical stories, two peaks were observed in the EEG spectrum, one at 4 Hz, that is, the syllable rate (p=0.0001, bootstrap, false discovery rate [FDR] corrected) and the other at 2 Hz, that is, the rate of disyllabic words (p=0.0001, bootstrap, FDR corrected). For nonmetrical stories, however, a single response peak was observed at 4 Hz (p=0.0001, bootstrap, FDR corrected), while no significant response peak was observed at 2 Hz (p=0.27, bootstrap, FDR corrected). A comparison of the responses to metrical and nonmetrical stories was performed, and a significant difference was observed at 2 Hz (p=0.0005, bootstrap, FDR corrected, *Figure 3A*) but not at 4 Hz (p=0.40, bootstrap, FDR corrected, *Figure 3B*). The response topography showed a centro-frontal distribution.

When participants watched a silent movie during story listening, however, a single response peak was observed at 4 Hz for both metrical (p=0.0002, bootstrap, FDR corrected) and nonmetrical stories (p=0.0002, bootstrap, FDR corrected) (*Figure 2C*). The response peak at 2 Hz was not significant for either kind of stories (p>0.074, bootstrap, FDR corrected). A comparison of the responses to metrical and nonmetrical stories did not find significant difference at either 2 Hz (p=0.22, bootstrap, FDR corrected, *Figure 3A*) or 4 Hz (p=0.39, bootstrap, FDR corrected, *Figure 3B*). Furthermore, the 2-Hz response was significantly stronger in the story comprehension task than in the movie watching task (p=0.0004, bootstrap, FDR corrected, *Figure 3A*). These results showed that cortical activity was synchronous to the word rhythm during attentive speech comprehension. When attention was diverted, however, the word-rate response was no longer detected.

## Neural encoding of words in natural spoken narratives

Next, we asked whether cortical activity was synchronous to disyllabic words in natural speech processing. The same set of stories used in the isochronous speech condition was read in a natural manner by a human speaker and presented to participants. The participants correctly answered 95 ± 4% and 97 ± 6% comprehension questions for metrical and nonmetrical stories, respectively. In natural speech, syllables were not produced at a constant rate (*Figure 1C*), and therefore the syllable and word responses were not frequency tagged. Nevertheless, we time warped the response to natural speech and made the syllable and word responses periodic. Specifically, the neural response to each syllable in natural speech was extracted and realigned to a constant 4-Hz rhythm using a convolution-based procedure (*Jin et al., 2018*; see Materials and methods for details). After time-warping analysis, cortical activity synchronous to the word onsets was expected to show a 2-Hz rhythm, the same as the response to the isochronous speech (*Figure 2E*).

The spectrum of the time-warped response was averaged over participants and EEG electrodes (*Figure 2E*). For metrical stories, two peaks were observed in spectrum of the time-warped response, one at 4 Hz (p=0.0002, bootstrap, FDR corrected) and the other at 2 Hz (p=0.0007, bootstrap, FDR corrected). For nonmetrical stories, however, a single response peak was observed at 4 Hz (p=0.0002, bootstrap, FDR corrected), while no significant response peak was observed at 2 Hz (p=0.37, bootstrap, FDR corrected). A comparison of the responses to metrical and nonmetrical stories was performed and found a significant difference at 2 Hz (p=0.036, bootstrap, FDR corrected, *Figure 3A*) but not at 4 Hz (p=0.09, bootstrap, FDR corrected, *Figure 3B*). These results demonstrated that cortical activity was synchronous to the word rhythm in natural spoken narratives during attentive speech comprehension.

## Neural responses to amplitude-modulated speech

To investigate potential interactions between neural encoding of linguistic units and relevant acoustic features, we amplitude modulated the isochronous speech at the word rate, that is, 2 Hz. The amplitude modulation (AM) was achieved by amplifying either σ1 or σ2 by a factor of 4 (*Figure 1D*), creating two conditions: σ1-amplified condition and σ2-amplified condition. Such 2-Hz AM provided an acoustic cue for the word rhythm. The speech was referred to as amplitude-modulated speech in the following. When listening to amplitude-modulated speech, the participants correctly answered 94 ± 12% and 97 ± 8% comprehension questions in the σ1- and σ2-amplified conditions, respectively.

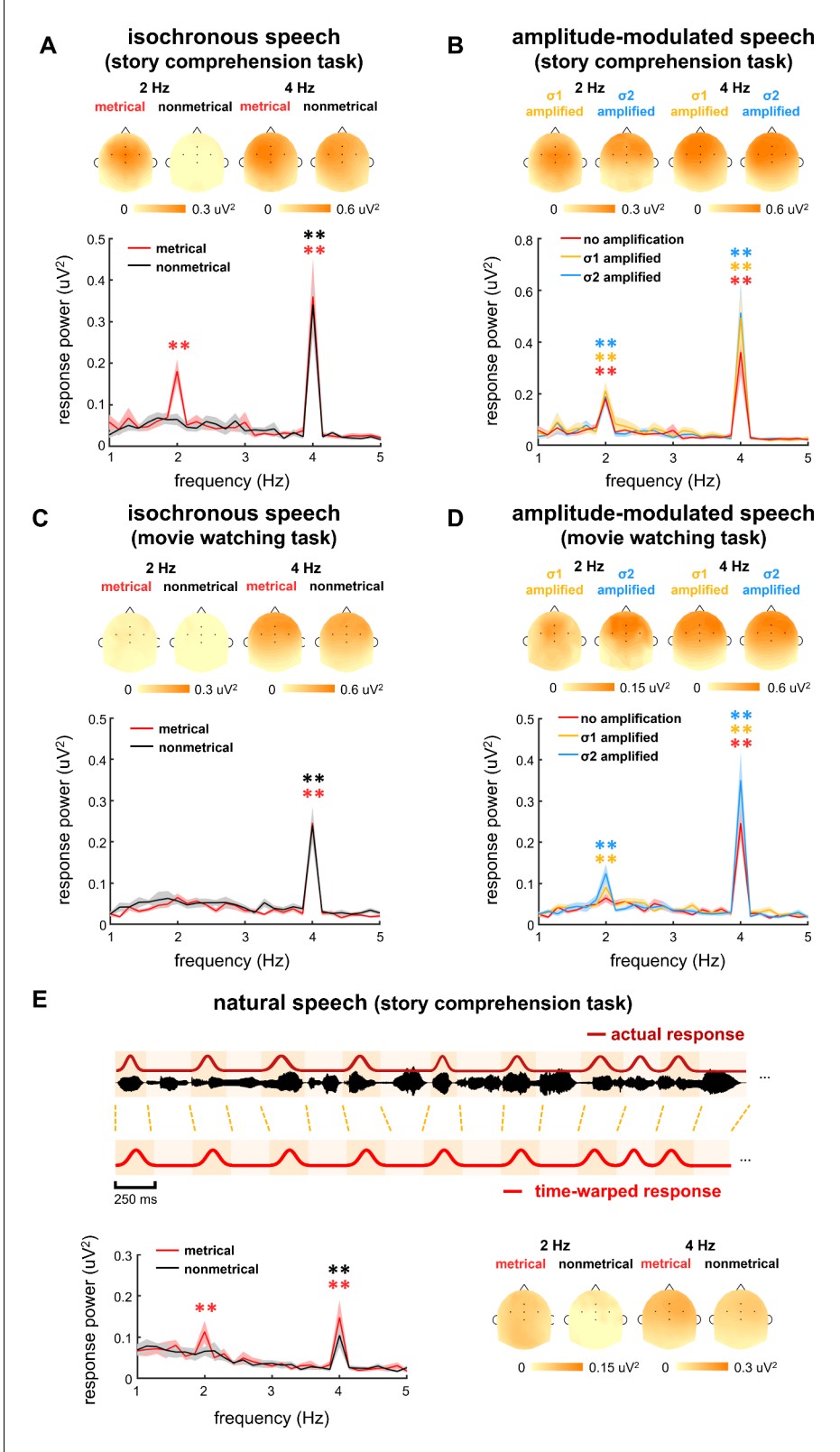

**Figure 2.** Electroencephalogram (EEG) response spectrum. Response spectrum is averaged over participants and EEG electrodes. The shaded area indicates one standard error of the mean (SEM) across participants. Stars indicate significantly higher power at 2 or 4 Hz than the power averaged over four neighboring frequency bins (two on each side). *p<0.05, **p<0.001 (bootstrap, false discovery rate [FDR] corrected). The color of the star is the same as the color of the spectrum. The topography on the top of each plot shows the distribution of response power at 2 Hz and 4 Hz. The five black

*Figure 2 continued on next page*

*Figure 2 continued*

dots in the topography indicate the position of electrodes FCz, Fz, Cz, FC3, and FC4. (A–D) Response spectrum for isochronous speech and amplitude-modulated speech during two tasks. To facilitate the comparison between stimuli, the red curves in panels **A** and **C** are repeated in panels **B** and **D**, respectively. (E) Response spectrum when the participants listen to natural speech. In this analysis, the response to natural speech is time warped to simulate the response to isochronous speech, and then transformed into the frequency-domain.

The online version of this article includes the following source data and figure supplement(s) for figure 2:

**Source data 1.** Preprocessed electroencephalogram (EEG) data recorded in Experiments 1–3.
**Figure supplement 1.** 2-Hz response power in individual electroencephalogram (EEG) electrodes and individual participants.

The EEG responses to amplitude-modulated speech are shown in *Figure 2B and D*. When participants attended to the speech, significant response peaks were observed at 2 Hz (σ1-amplified: p=0.0001, and σ2-amplified: p=0.0001, bootstrap, FDR corrected, *Figure 2D*). A comparison between the responses to isochronous speech and amplitude-modulated speech found that AM did not significantly influence the power of the 2-Hz neural response (σ1-amplified vs. σ2-amplified: p=0.56; σ1-amplified vs. isochronous: p=0.38, σ2-amplified vs. isochronous: p=0.44, bootstrap, FDR corrected, *Figure 3A*). These results showed that the 2-Hz response power was not significantly influenced by the 2- Hz AM during attentive speech comprehension.

Another group of participants passively listened to amplitude-modulated speech while attending to a silent movie. In their EEG responses, significant response peaks were also observed at 2 Hz (σ1-amplified: p=0.0001, and σ2-amplified: p=0.0001, bootstrap, FDR corrected, *Figure 2D*). A comparison between the responses to isochronous speech and amplitude-modulated speech showed stronger 2-Hz response in the processing of amplitude-modulated speech than isochronous speech (σ1-amplified vs. isochronous: p=0.021, σ2-amplified vs. isochronous: p=0.0042, bootstrap, FDR corrected, *Figure 3A*). No significant difference was found between responses to σ1-amplified and σ2-amplified speech (p=0.069, bootstrap, FDR corrected, *Figure 3A*). Therefore, when attention was diverted, the 2-Hz response power was significantly increased by the 2-Hz AM.

The Fourier transform decomposes an arbitrary signal into sinusoids and each complex-valued Fourier coefficient captures the magnitude and phase of a sinusoid. The power spectrum reflects the response magnitude but ignores the response phase. The phase difference between the 2-Hz responses in the σ1- and σ2-amplified conditions, however, carried important information about whether the neural response was synchronous to the word onsets or amplified syllables: Neural activity synchronous to amplified syllables showed a 250 ms time lag between the σ1- and σ2-amplified conditions (*Figure 1D*), which corresponded to a 180˚ phase difference between conditions at 2 Hz. Neural activity synchronous to the word onsets, however, should be identical in the σ1- and σ2-amplified conditions.

The 2-Hz response phase difference between the σ1- and σ2-amplified conditions is shown in *Figure 3C*. The phase difference, when averaged across participants and EEG electrodes, was 41˚ (95% confidence interval: −25–91˚) during the story comprehension task and increased to 132˚ (95% confidence interval: 102–164˚) during the movie watching task (see *Figure 3—figure supplement 1* for response phase in individual conditions).

## Separate the responses to words and word-rate AM

In the previous section, we analyzed the difference in 2-Hz response phase difference between the σ1- and σ2-amplified conditions. A 0˚ phase difference indicated that the 2-Hz response was only synchronous to word onsets, while a 180˚ phase difference indicated that the 2-Hz response was only synchronous to amplified syllables. A phase difference between 0˚ and 180˚ indicated that neural response synchronous to word onsets and neural response synchronous to amplified syllables both exist, but the phase analysis could not reveal the strength of the two response components. Therefore, in the following, we extracted the neural responses to words and the 2-Hz AM by averaging the responses across the σ1- and σ2-amplified conditions in different manners.

To extract the neural response to words, we averaged the neural response waveforms across the σ1- and σ2-amplified conditions. The word onsets were aligned in these two conditions and therefore neural activity synchronous to the word onsets was preserved in the average. In contrast, cortical activity synchronous to the amplified syllables exhibited a 250 ms time lag between the σ1- and

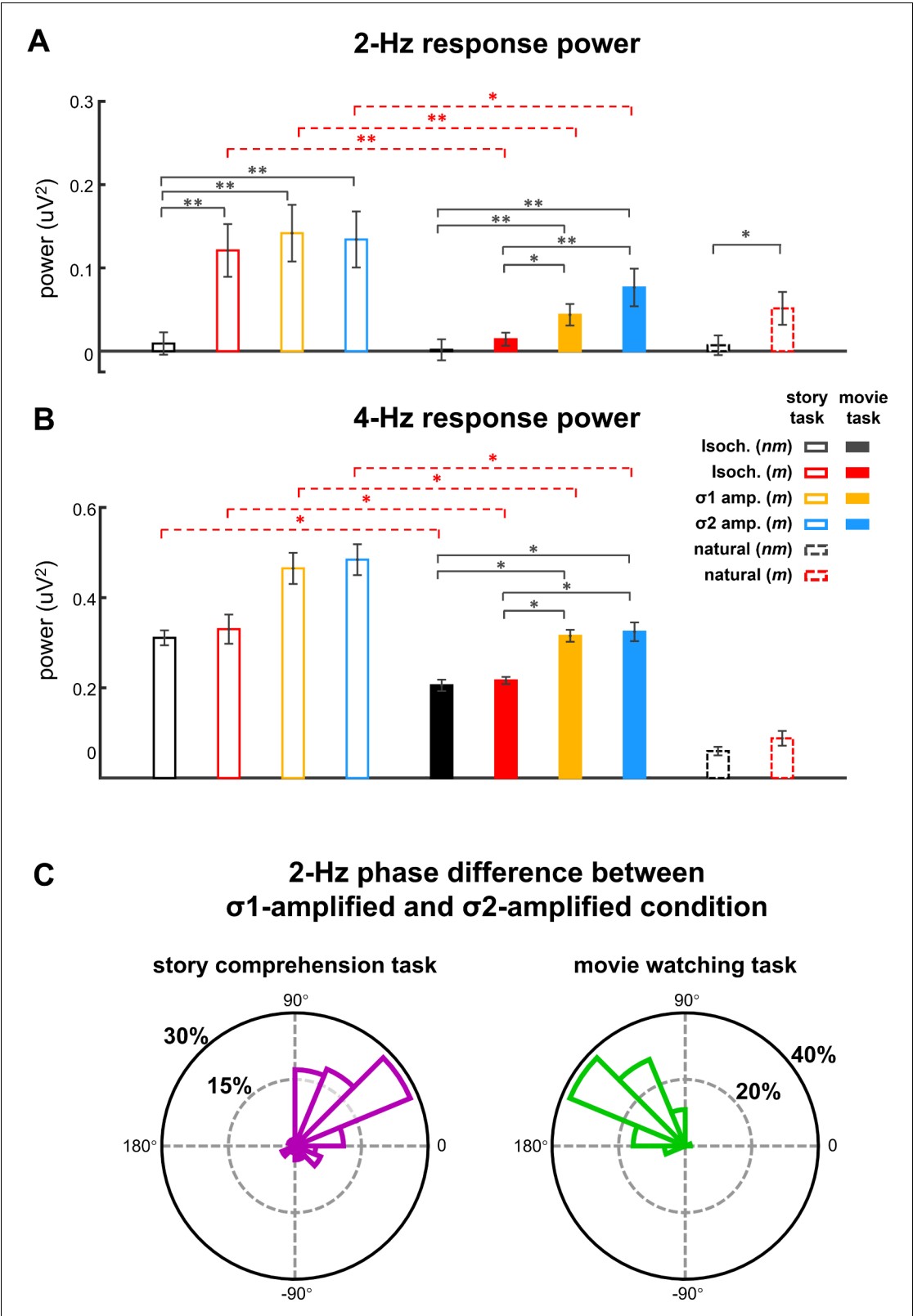

**Figure 3.** Response power and phase. (**A and B**) Response power at 2 and 4 Hz. Color of the bars indicates the stimulus. Black stars indicate significant differences between different types of speech stimuli while red stars indicate significant differences between tasks. *p<0.05, **p<0.01 (bootstrap, false discovery rate [FDR] corrected). Throughout the manuscript, in all bar graphs of response power, the response power at a target frequency is subtracted by the power averaged over four neighboring frequency bins (two on each side) to reduce the influence of background neural activity. (**C**)

*Figure 3 continued on next page*

*Figure 3 continued*

The difference in 2-Hz response phase between the σ1- and σ2-amplified conditions at 2 Hz. The phase difference is averaged across participants, and the polar histogram shows the distribution of phase difference across 64 electrodes.

The online version of this article includes the following source data and figure supplement(s) for figure 3:

**Source data 1.** Preprocessed EEG data recorded in Experiment 1-3.

**Figure supplement 1.** 2-Hz response phase in individual electroencephalogram (EEG) electrodes.

σ2-amplified conditions. Therefore, the 2-Hz response synchronous to the amplified syllables was 180° out of phase between the two conditions, and got canceled in the average across conditions. In sum, the average over the σ1- and σ2-amplified conditions preserved the response to words but canceled the response to amplified syllables (*Figure 4A*). This average was referred to as the word response in the following.

For the word response, a significant response peak was observed at 2 Hz during both story comprehension task (p=0.0001, bootstrap, FDR corrected) and movie watching task (p=0.011, bootstrap, FDR corrected, *Figure 4B*). The response power was significantly stronger in the story comprehension task than that in the movie watching task (p=0.0008, bootstrap, FDR corrected, *Figure 4E*). During the movie watching task, a significant 2-Hz word response was observed when amplitude-modulated speech was presented (p=0.011, bootstrap, FDR corrected, *Figure 4B*), while no significant 2-Hz word response was observed when isochronous speech was presented (p=0.074, bootstrap, FDR corrected, *Figure 2C*). This finding indicated that the 2-Hz AM facilitated word processing during passive listening.

Furthermore, during the story comprehension task, the power of the 2-Hz word response did not significantly differ between amplitude-modulated speech and isochronous speech (p=0.69, bootstrap, FDR corrected), suggesting that the 2-Hz AM did not significantly modulate the 2-Hz word response during attentive speech comprehension.

To extract the neural response to 2-Hz AM, we first aligned the responses in the σ1- and σ2-amplified conditions by adding a delay of 250 ms to the response in the σ1-amplified condition. We then averaged the response waveforms across conditions (*Figure 4C*). After the delay was added to the response to σ1-amplified speech, the 2-Hz AM was identical between the σ1- and σ2-amplified conditions. Therefore, the average across conditions preserved the response to 2-Hz AM while canceling the response to words. The averaged response was referred to as the AM response in the following.

For the AM response, significant 2-Hz response peaks were observed during both the story comprehension task (p=0.0021, bootstrap, FDR corrected) and the movie watching task (p=0.0001, bootstrap, FDR corrected). The 2-Hz response power did not significantly differ between tasks (p=0.39, bootstrap, FDR corrected), suggesting that the 2-Hz AM response was not strongly enhanced when participants attended to speech.

During the story comprehension task, the 2-Hz response power averaged across electrodes was significantly stronger for the word response than the AM response (p=0.021, bootstrap, FDR corrected, *Figure 4E*). During the movie watching task, however, the reverse was true, that is, the 2-Hz word response was significantly weaker than the 2-Hz AM response (p=0.013, bootstrap, FDR corrected, *Figure 4E*).

An analysis of individual electrodes showed the 2-Hz word response was significantly stronger than the 2-Hz AM response in centro-frontal electrodes during the story comprehension task (p<0.05, bootstrap, FDR corrected, *Figure 4F*, left). To further compare the spatial distribution of 2-Hz word and AM responses, on top of their power difference, we normalized the response topography by dividing its maximum value. Significant difference was found in the normalized topography between the 2-Hz word and AM responses in temporal electrodes (p<0.05, bootstrap, FDR corrected, *Figure 4F*, right). These results suggested that the 2-Hz word and AM responses had distinct neural sources.

## Neural responses to amplitude-modulated speech: a replication

The responses to amplitude-modulated speech suggested that the brain encoded words and acoustic cues related to words in a different manner. Furthermore, the word-related acoustic cues seemed

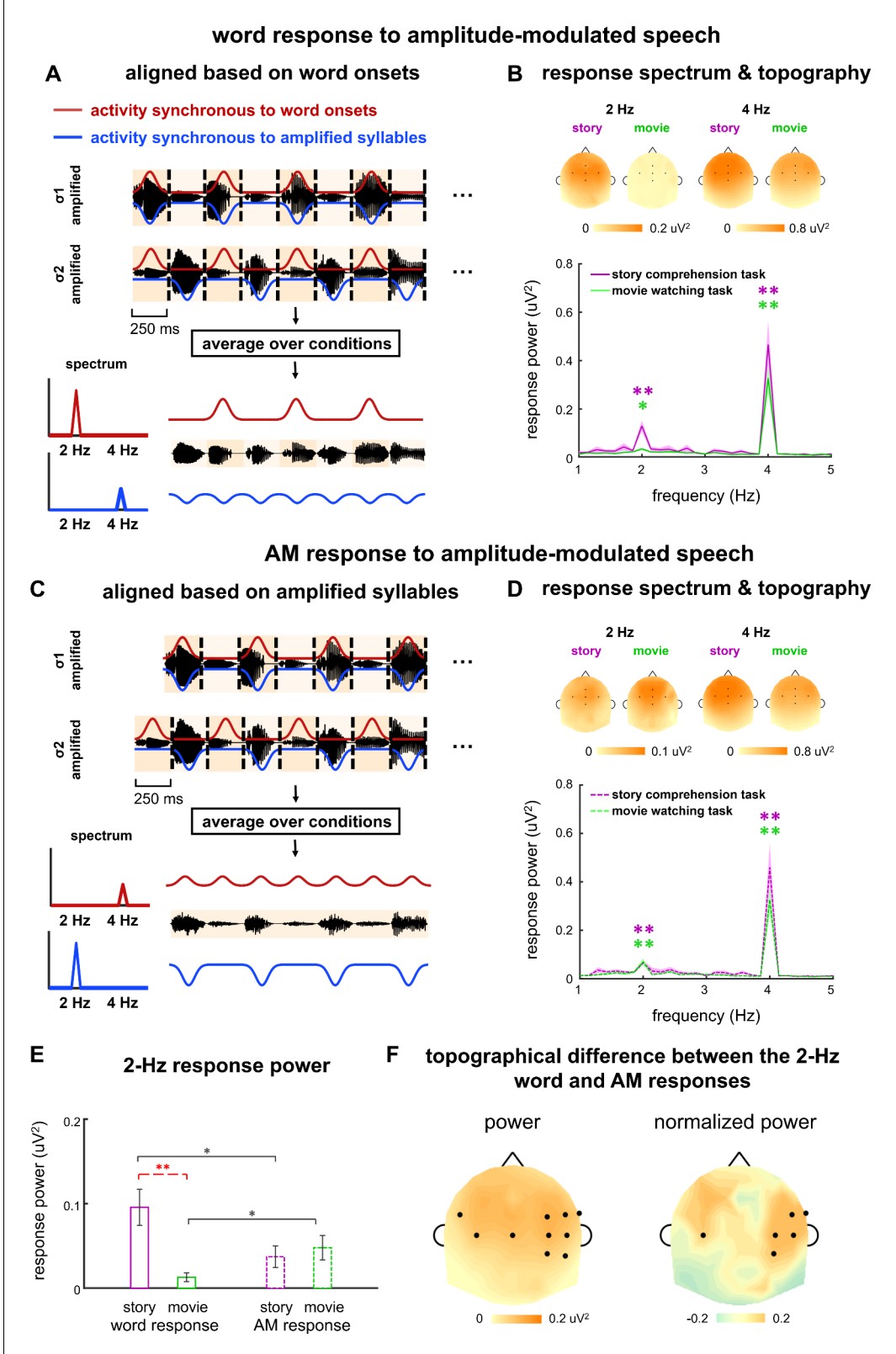

**Figure 4.** Word and amplitude modulation (AM) responses to amplitude-modulated speech. (A–D) Neural responses in σ1- and σ2-amplified conditions are aligned based on either word onsets (AB) or amplified syllables (CD), and averaged to extract the 2-Hz response synchronous to words or AM, respectively. Panels **A** and **C** illustrate the procedure. The red and blue curves illustrate the response components synchronous to word onsets and the AM respectively, which are mixed in the electroencephalogram (EEG) measurement and shown separately for illustrative purposes. The

*Figure 4 continued on next page*

*Figure 4 continued*

spectrum and topography in panels B and D are shown the same way as they are shown in *Figure 2*. *p<0.05, **p<0.001 (bootstrap, false discovery rate [FDR] corrected). (E) Response power at 2 Hz. Black stars indicate significant differences between the word and AM responses, while red stars indicate a significant difference between tasks. *p<0.05, **p<0.01 (bootstrap, FDR corrected). (F) The left panel shows the power difference between the word and AM responses in single electrodes. The right panel shows the difference in normalized topography, that is, topography divided by its maximal value. Black dots indicate electrodes showing a significant difference between the word and AM responses (p<0.05, bootstrap, FDR corrected).

The online version of this article includes the following source data for figure 4:

**Source data 1.** Preprocessed electroencephalogram (EEG) data recorded in Experiments 1–3.

to facilitate word processing during passive listening, but not significantly enhance the word response during attentive listening. To further validate these findings, we conducted a replication experiment to measure the neural response to amplitude-modulated speech in a separate group of participants. These participants first performed the movie watching task and then the story comprehension task. During the story comprehension task, the participants correctly answered 96 ± 8% and 95 ± 9% questions in the σ1-amplified and σ2-amplified conditions, respectively.

We first analyzed the spectrum of response to σ1- and σ2-amplified speech. Consistent with previous results, a significant 2-Hz response peak was observed whether the participants attended to the speech (σ1-amplified: p=0.0001, and σ2-amplified: p=0.0001, bootstrap, FDR corrected) or movie (σ1-amplified: p=0.0001, and σ2-amplified: p=0.0004, bootstrap, FDR corrected, *Figure 5—figure supplement 1A and B*). When averaged across participants and electrodes, the 2-Hz response phase difference between the σ1- and σ2-amplified conditions was 7° (95% confidence interval: −26–47°) during the story comprehension task and 96° (95% confidence interval: 55–143°) during the movie watching task (*Figure 5—figure supplement 1C and D*).

We then separately extracted the word and AM responses following the procedures described in *Figure 4A and C*. Similar to previous results, in the spectrum (*Figure 5A and B*), a significant 2-Hz word response was observed during both tasks (story comprehension task: p=0.0001, bootstrap; movie watching task: p=0.0003, bootstrap, FDR corrected), and a significant 2-Hz AM response was also observed during both tasks (story comprehension task: p=0.0001, bootstrap; movie watching task: p=0.0001, bootstrap, FDR corrected). Furthermore, the 2-Hz word response exhibited significantly stronger power than the 2-Hz AM response during the story comprehension task (p=0.0036, bootstrap, FDR corrected, *Figure 5C*), and the two responses showed distinct topographical distribution (*Figure 5D*). In sum, the results obtained in the replication experiment were consistent with those from the original experiment. A follow-up comparison of the results from the original and replication experiments suggested that the 2-Hz word response exhibited significantly stronger power during the movie watching task in the replication experiment than the original experiment (p=0.0008, bootstrap, FDR corrected).

## Time course of EEG responses to words

The event-related potential (ERP) responses evoked by σ1 and σ2 are separately shown in *Figure 6*. The ERP analysis was restricted to disyllabic words so that the responses to σ1 and σ2 represented the responses to the first and second syllables of disyllabic words respectively. When participants attended to speech, the ERP responses to σ1 and σ2 showed significant differences for both isochronous (*Figure 6A*) and natural speech (*Figure 6B*). When participants watched a silent movie, a smaller difference was also observed between the ERP responses to σ1 and σ2 (*Figure 6A*). The topography of the ERP difference showed a centro-frontal distribution. For isochronous speech, the ERP latency could not be unambiguously interpreted for isochronous speech, given that the stimulus was strictly periodic. For natural speech, the ERP responses to σ1 and σ2 differed in a time window around 300–500 ms.

The ERP results for amplitude-modulated speech are shown in *Figure 6C and D*. When participants attended to speech, a difference between the ERP responses to σ1 and σ2 was observed in both σ1- and σ2-amplified conditions (*Figure 6C*). During passive listening, however, a significant ERP difference was observed near the onset of the amplified syllable. These results were consistent with the results in the replication experiment (*Figure 6—figure supplement 1*).

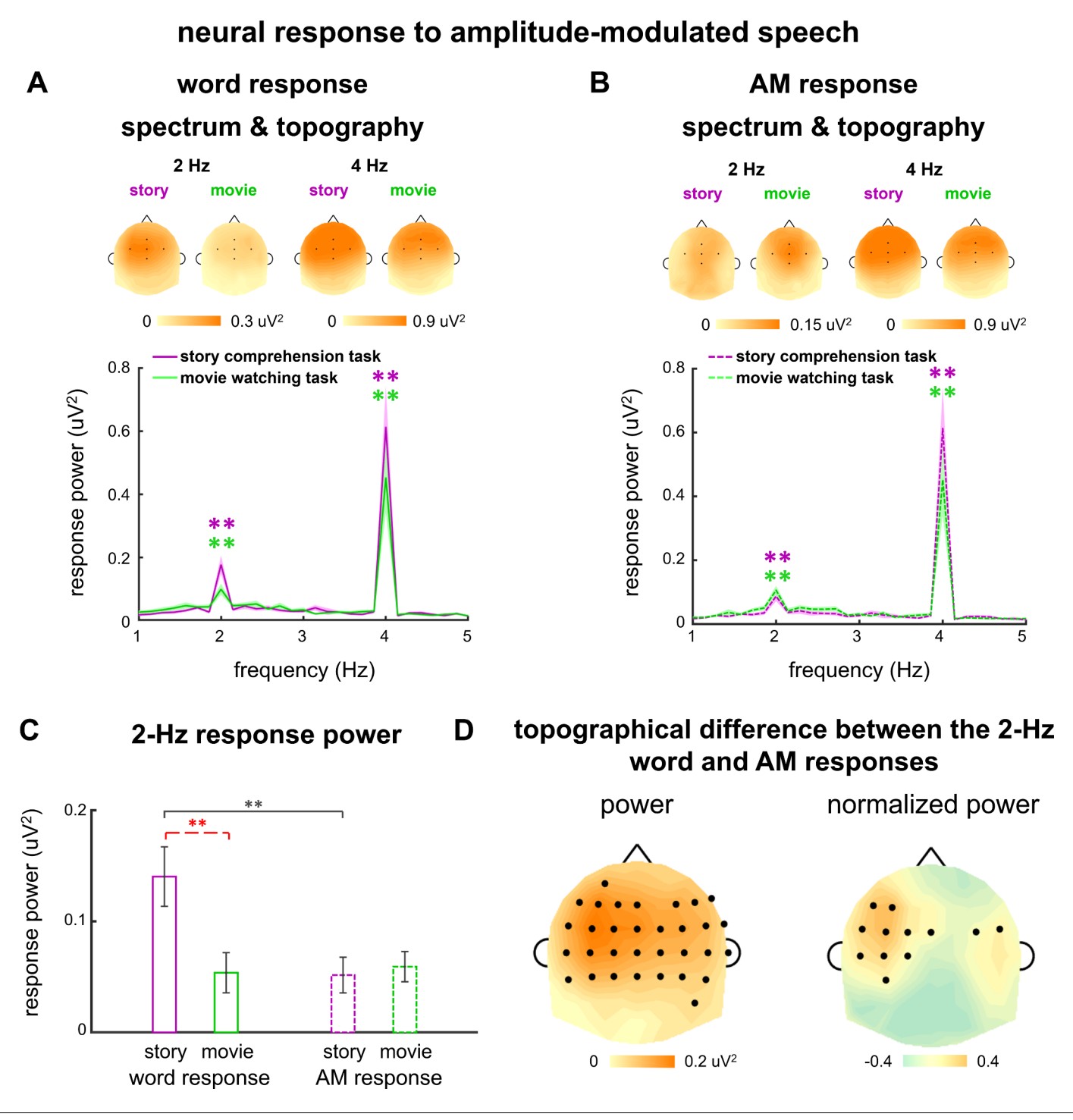

**Figure 5.** Replication of the neural response to amplitude-modulated speech. (**A and B**) Spectrum and topography for the word response (**A**) and amplitude modulation (AM) response (**B**). Colored stars indicate frequency bins with stronger power than the power averaged over four neighboring frequency bins (two on each side). *p<0.05, **p<0.001 (bootstrap, false discovery rate [FDR] corrected). The topography on the top of each plot shows the distribution of response power at 2 Hz and 4 Hz. (**C**) Response power at 2 Hz. Black stars indicate significant differences between the word and (AM) responses, while red stars indicate a significant difference between tasks. *p<0.05, **p<0.01 (bootstrap, FDR corrected). (**D**) Power difference between the word and AM responses in individual electrodes are shown in the left panel. To further illustrate the difference in topographical distribution instead of the response power, each response topography is normalized by dividing its maximum value. The difference in the normalized topography is shown in the right panel. Black dots indicate electrodes showing a significant difference between the word and AM responses (p<0.05, bootstrap, FDR corrected).

*Figure 5 continued on next page*

*Figure 5 continued*

The online version of this article includes the following source data and figure supplement(s) for figure 5:

**Source data 1.** Preprocessed electroencephalogram (EEG) data recorded in Experiment 4.

**Figure supplement 1.** Electroencephalogram (EEG) response spectrum and 2-Hz phase difference in the replication experiment.

## Discussion

Speech comprehension is a complex process involving multiple stages, for example, encoding of acoustic features, extraction of phonetic features, and processing of higher-level linguistic units such as words, phrases, and sentences. Here, we investigate how low-frequency cortical activity encodes linguistic units and related acoustic features. When participants naturally listen to spoken narratives, we observe that cortical activity is synchronous to the rhythm of spoken words. The word synchronous response is observed whether participants listen to natural speech or synthesized isochronous speech that removes word-related acoustic cues. Furthermore, when introducing an AM cue to the word rhythm, neural responses to words and the AM cue are both observed and they show different spatial distribution. In addition, when participants are engaged in a story comprehension task, the word response exhibits stronger power than the AM response. The AM cue does not clearly modulate the word response during story comprehension task (*Figure 2B*), but can facilitate word processing when attention is diverted: The word response is not detected for isochronous speech (*Figure 2C*), but is detected for amplitude-modulated speech during passive listening (*Figures 4B* and *5A*). In sum, these results show that both top-down linguistic knowledge and bottom-up acoustic cues separately contribute to word synchronous neural responses.

### Neural encoding of linguistic units in natural speech

In speech, linguistic information is organized through a hierarchy of units, including phonemes, syllables, morphemes, words, phrases, sentences, and discourses. These units span a broad range of time scales, from tens of milliseconds for phonemes to a couple of seconds for sentences, and even longer for discourses. It is a challenging question to understand how the brain represents the hierarchy of linguistic units, and it is an appealing hypothesis that each level of linguistic unit is encoded by cortical activity on the relevant time scale (*Ding et al., 2016a*; *Doumas and Martin, 2016*; *Giraud and Poeppel, 2012*; *Goswami, 2019*; *Keitel et al., 2018*; *Kiebel et al., 2008*; *Meyer and Gumbert, 2018*). Previous fMRI studies have suggested that neural processing at different levels, for example, syllables, words, and sentences, engages different cortical networks (*Blank and Fedorenko, 2020*; *Hasson et al., 2008*; *Lerner et al., 2011*). Magnetoencephalography (MEG)/ Electroencephalogram (EEG) studies have found reliable delta- and theta-band neural responses that are synchronous to speech (*Ding and Simon, 2012*; *Luo and Poeppel, 2007*; *Peelle et al., 2013*), and the time scales of such activity are consistent with the time scales of syllables and larger linguistic units.

Nevertheless, it remains unclear whether these MEG/EEG responses directly reflect neural encoding of hierarchical linguistic units, or simply encode acoustic features associated with these units (*Daube et al., 2019*; *Kösem and van Wassenhove, 2017*). On one hand, neural tracking of sound envelope is reliably observed in the absence of speech comprehension, for example, when participants listen to unintelligible speech (*Howard and Poeppel, 2010*; *Zoefel and VanRullen, 2016*) and non-speech sound (*Lalor et al., 2009*; *Wang et al., 2012*). The envelope tracking response is even weaker for sentences composed of real words than sentences composed of pseudowords (*Mai et al., 2016*), and is weaker for the native language than an unfamiliar language (*Zou et al., 2019*). Neural tracking of speech envelope can also be observed in animal primary auditory cortex (*Ding et al., 2016b*). Furthermore, a recent study shows that low-frequency cortical activity cannot reflect the perception of an ambiguous syllable sequence, for example, whether repetitions of a syllable is perceived as 'flyflyfly' or 'lifelifelife' (*Kösem et al., 2016*).

On the other hand, cortical activity synchronous to linguistic units, such as words and phrases, can be observed using well-controlled synthesized speech that removes relevant acoustic cues (*Ding et al., 2016a*; *Jin et al., 2018*; *Makov et al., 2017*). These studies, however, usually present semantically unrelated words or sentences at a constant pace, which creates a salient rhythm easily noticeable for listeners. In contrast, in the current study, we presented semantically coherent stories.

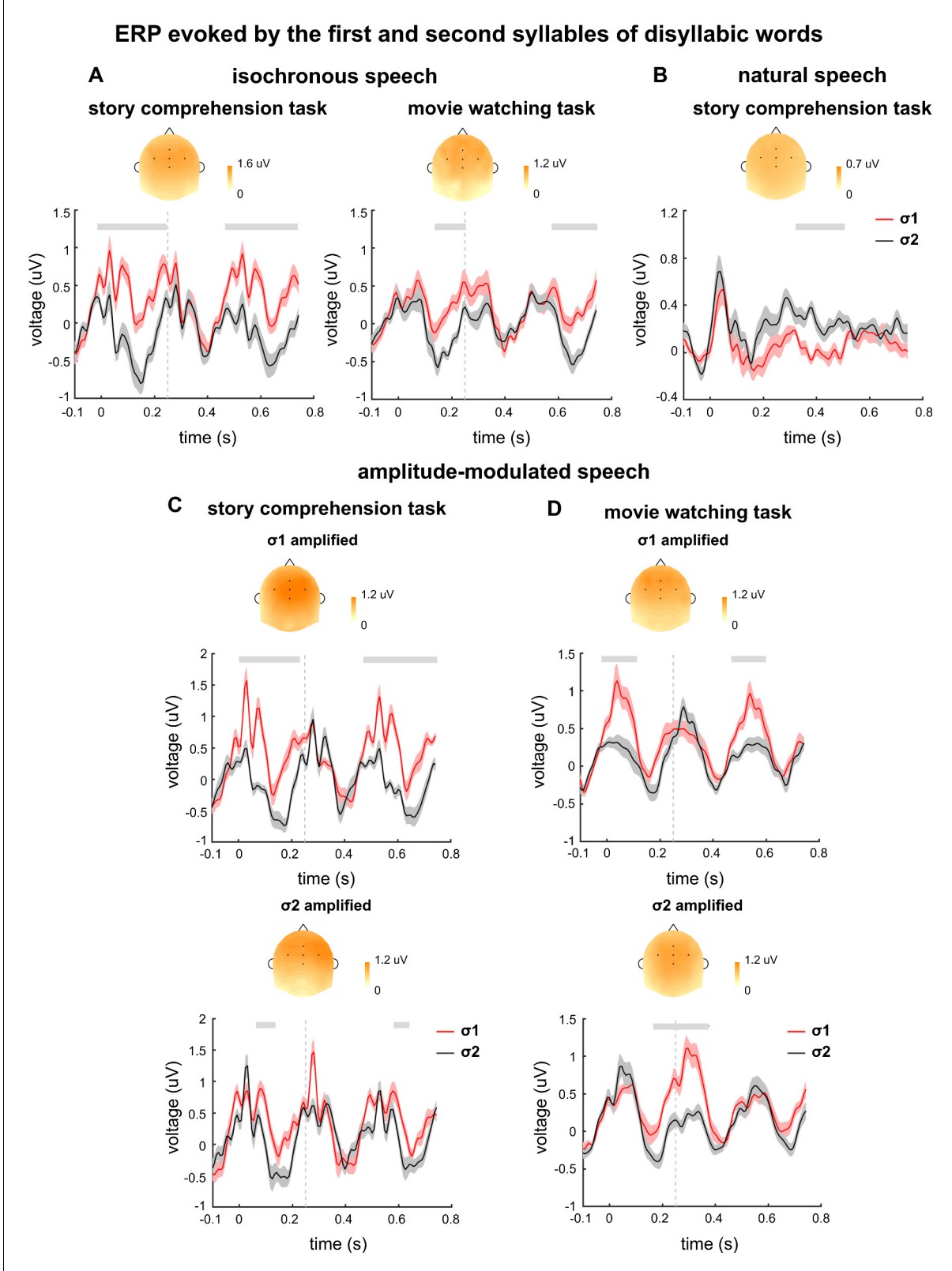

**Figure 6.** Event-related potential (ERP) responses evoked by disyllabic words. The ERP responses evoked by σ1 and σ2 are shown in red and black, respectively. The ERP response is averaged over participants and electrodes. The shaded area indicates 1 SEM across participants. The gray lines on top denote the time intervals in which the two responses are significantly different from each other (p<0.05, cluster-based permutation test). The
*Figure 6 continued on next page*

*Figure 6 continued*

topography on top is averaged over all time intervals showing a significant difference between the two responses in each plot. Time 0 indicates syllable onset.

The online version of this article includes the following source data and figure supplement(s) for figure 6:

**Source data 1.** Preprocessed electroencephalogram (EEG) data recorded in Experiments 1–3.

**Figure supplement 1.** Event-related potential (ERP) responses evoked by disyllabic words in the replication experiment.

It was found that for both synthesized isochronous speech and natural speech, cortical activity was synchronous to multi-syllabic words in metrical stories when participants were engaged in a story comprehension task. Furthermore, few listeners reported noticing any difference between the metrical and nonmetrical stories (see Supplementary materials for details), suggesting that the word rhythm was not salient and was barely noticeable to the listeners. Therefore, the word response is likely to reflect implicit word processing instead of the perception of an explicit rhythm.

A comparison of the responses to natural speech and isochronous speech showed that responses to word and syllable were weaker for natural speech, suggesting that strict periodicity in the stimulus could indeed boost rhythmic neural entrainment. Although the current study and previous studies (*Ding et al., 2018*; *Makov et al., 2017*) observe a word-rate neural response, the study conducted by *Kösem et al., 2016* does not report observable neural activity synchronous to perceived word rhythm. A potential explanation for the mixed results is that *Kösem et al., 2016* repeat the same word in each trial while the other studies present a large variety of words with no immediate repetitions in the stimuli. Therefore, it is possible that low-frequency word-rate neural response more strongly reflects neural processing of novel words, instead of the perception of a steady rhythm (see also *Ostarek et al., 2020*).

## Mental processes reflected in neural activity synchronous to linguistic units

It remains elusive what kind of mental representations are reflected by cortical responses synchronous to linguistic units. For example, the response may reflect the phonological, syntactic, or semantic aspect of a perceived linguistic unit and it is difficult to tease apart these factors. Even if a sequence of sentences is constructed with independently synthesized monosyllabic words, the sequence does not sound like a stream of individual syllables delivered at a constant pace. Instead, listeners can clearly perceive each sentence as a prosodic unit. In this case, mental construction of sentences is driven by syntactic processing. Nevertheless, as long as the mental representation of a sentence is formed, it also has an associated phonological aspect. Previous psycholinguistic studies have already demonstrated that syntax has a significant impact on prosody perception (*Buxó-Lugo and Watson, 2016*; *Garrett et al., 1966*).

It is also possible that neural activity synchronous to linguistic units reflect more general cognitive processes that are engaged during linguistic processing. For example, within a word, later syllables are more predictable that earlier syllables. Therefore, neural processing associated with temporal prediction (*Breska and Deouell, 2017*; *Lakatos et al., 2013*; *Stefanics et al., 2010*) may appear to be synchronous to the perceived linguistic units. However, it has been demonstrated that when the predictability of syllables is controlled as a constant, cortical activity can still synchronize to perceived artificial words, suggesting that temporal prediction is not the only factor driving low-frequency neural activity either (*Ding et al., 2016a*). Nevertheless, in natural speech processing, temporal prediction may inevitably influence the low-frequency response. Similarly, temporal attention is known to affect low-frequency activity and attention certainly varies during speech perception (*Astheimer and Sanders, 2009*; *Jin et al., 2018*; *Sanders and Neville, 2003*). In addition, low-frequency neural activity has also been suggested to reflect the perception of high-level rhythms (*Nozaradan et al., 2011*) and general sequence chunking (*Jin et al., 2020*).

Since multiple factors can drive low-frequency neural responses, the low-frequency response to natural speech is likely to be a mixture of multiple components, including, for example, auditory responses to acoustic prosodic features, neural activity related to temporal prediction and temporal attention, and neural activity encoding phonological, syntactic, or semantic information. These processes are closely coupled and one process can trigger other processes. Here we do not think the

word-rate response exclusively reflects a single process. It may well consist of a mixture of multiple response components, or provide a reference signal to bind together mental representations of multiple dimensions of the same word. Similar to the binding of color and shape information in the perception of a visual object (*Treisman, 1998*), the perception of a word requires the binding of, for example, phonological, syntactic, and semantic representations. It has been suggested that temporal coherence between neural responses to different features provides a strong cue to bind these features into a perceived object (*Shamma et al., 2011*). It is possible that the word-rate response reflects the temporal coherence between distinct mental representations of each word and functionally relates to the binding of these representations.

In speech processing, multiple factors contribute to the word response and these factors interact. For example, the current study suggested that prosodic cues such as AM had a facilitative effect on word processing: It was observed that, during passive listening, a word response was observed for amplitude-modulated speech but not for isochronous speech. In addition, the word response to amplitude-modulated speech during passive listening exhibited stronger power in the replication experiment that only presented amplitude-modulated speech, compared with the results obtained in the original experiment that presented amplitude-modulated speech with isochronous speech in a mixed manner. Consistent with this finding, a larger number of participants reported the noticing of stories during passive listening in the replication experiment (see Materials and methods). All these results suggest that the 2-Hz AM, which provides an acoustic cue for the word rhythm, facilitates word processing during passive listening. This result is consistent with the idea that prosodic cues have a facilitative effect on speech comprehension (*Frazier et al., 2006*; *Ghitza, 2017*; *Ghitza, 2020*; *Giraud and Poeppel, 2012*).

Finally, it should be mentioned that we employed AM to manipulate the speech envelope, given that the speech envelope is one of the strongest cues to drive stimulus-synchronous cortical response. However, the AM is only a weak prosodic cue compared with other variables such as timing and pitch contour (*Shen, 1993*). Furthermore, stress is strongly modulated by context and does not affect word recognition in Chinese (*Duanmu, 2001*). Future studies are needed to characterize the modulation of language processing by different prosodic cues and investigate the modulatory effect across different languages.

## Attention modulation of cortical speech responses

It has been widely reported that cortical tracking of speech is strongly modulated by attention. Most previous studies demonstrate that in a complex auditory scene consisting of two speakers, attention can selectively enhance neural tracking of attended speech (*Ding and Simon, 2012*; *Zion Golumbic et al., 2013*). These results strongly suggest that the auditory cortex can parse a complex auditory scene into auditory objects, for example, speakers, and separately represent each auditory object (*Shamma et al., 2011*; *Shinn-Cunningham, 2008*). When only one speech stream was presented, cross-modal attention can also modulate neural tracking of the speech envelope, but the effect is much weaker (*Ding et al., 2018*; *Kong et al., 2014*).

Consistent with previous findings, in the current study, the 4-Hz syllable response was also enhanced by cross-modal attention (*Figure 3B*). The 2-Hz AM response power, however, was not significantly modulated by cross-modal attention (*Figures 4D* and *5B*), suggesting that attention did not uniformly enhance the processing of all features within the same speech stream. Given that the 2-Hz AM does not carry linguistic information, the result suggests that attention selectively enhances speech features relevant to speech comprehension. This result extends previous findings by showing that attention can differentially modulate different features within a speech stream.

## Time course of the neural response to words

The neurophysiological processes underlying speech perception has been extensively studied using ERPs (*Friederici, 2002*). Early ERP components, such as the N1, mostly reflect auditory encoding of acoustic features, while late components can reflect higher-level processing of lexical, semantic, or syntactic processing (*Friederici, 2002*; *Friederici, 2012*; *Sanders and Neville, 2003*). In the current study, for isochronous speech, response latency cannot be uniquely determined: Given that syllables are presented at a constant rate of 4 Hz, a response with latency T cannot be distinguished from responses with latency T ± 250 ms. The periodic pattern in the stimulus design enables accurate

prediction of its timing in the brain, and therefore the responses observed could turn out to be predictive instead of reactive.

In natural speech, the responses to the two syllables in a disyllabic word differ in a late latency window of about 400 ms. This component is consistent with the latency of the N400 response, which can be observed when listening to either individual words or continuous speech (*Broderick et al., 2018*; *Kutas and Federmeier, 2011*; *Kutas and Hillyard, 1980*; *Pylkkänen and Marantz, 2003*; *Pylkkänen et al., 2002*). A previous study on the neural responses to naturally spoken sentences has also shown that the initial syllable of an English word elicits larger N1 and N200–300 components than the word-medial syllable (*Sanders and Neville, 2003*). A recent study also suggests that the word onset in natural speech elicits a response at ~100 ms latency (*Brodbeck et al., 2018*). The current study, however, does not observe this early effect, and language difference might offer a potential explanation: In Chinese, a syllable generally equals a morpheme, while in English single syllables do not carry meaning in most cases. The 400 ms latency response observed in the current study is consistent with the hypothesis that the N400 is related to lexical processing (*Friederici, 2002*; *Kutas and Federmeier, 2011*). Besides, it is also possible that the second syllable in a disyllabic word elicits weaker N400 since it is more predictable than the first syllable (*Kuperberg et al., 2020*; *Lau et al., 2008*).

The difference between the ERPs evoked by the first and second syllables in disyllabic words was amplified by attention (*Figure 6A*). Furthermore, the ERP difference remained significant when participants' attention was diverted away from the speech in the movie watching task, while the 2-Hz response in the power spectrum was no longer significant (*Figure 2C*). These results are similar to the findings of a previous study (*Ding et al., 2018*), in which no word-rate response peak in the EEG spectrum is observed in contrast to coherent word-rate response phase across participants in unattended listening task. Taken together, these results suggest that word-level processing occurs during the movie watching task, but the word-tracking response is rather weak.

In sum, the current study suggests that bottom-up acoustic cues and top-down linguistic knowledge separately contribute to neural construction of linguistic units in the processing of spoken narratives.

## Materials and methods

### Participants
Sixty-eight participants (20–29 years old, mean age, 22.6 years; 37 females) took part in the EEG experiments. Thirty-four participants (19–26 years old, mean age, 22.5 years; 17 females) took part in a behavioral test to assess the naturalness of the stimuli. All participants were right-handed native Mandarin speakers, with no self-reported hearing loss or neurological disorders. The experimental procedures were approved by the Research Ethics Committee of the College of Medicine, Zhejiang University (2019–047). All participants provided written informed consent prior to the experiment and were paid.

### Stories
Twenty-eight short stories were constructed for the study. The stories were unrelated in terms of content and ranged from 81 to 143 in word count (107 words on average). In 21 stories, word onset was metrically organized and every other syllable was a word onset, and these stories were referred to as metrical stories (*Figure 1A*). In the other seven stories, word onset was not metrically organized and these stories were referred to as nonmetrical stories (*Figure 1B*). In an ideal design, metrical stories should be constructed solely with disyllabic words to form a constant disyllabic word rhythm. Nevertheless, since it was difficult to construct such materials, the stories were constructed with disyllabic words and pairs of monosyllabic words. In other words, in between two disyllabic words, there must be an even number of monosyllabic words. After the stories were composed, the word boundaries within the stories were further parsed based on a Natural Language Processing (NLP) algorithm (*Zhang and Shang, 2019*). The parsing result confirmed that every other syllable in the story (referred to as σ1 in *Figure 1A*) was the onset of a word. For the other syllables (referred to as σ2 in *Figure 1A*), 77% was the second syllable of a disyllabic word while 23% was a monosyllabic word.

Similarly, nonmetrical stories used as a control condition were composed with sentences with an even number of syllables. Nevertheless, no specific constraint was applied to word duration, and the odd terms in the syllable sequence were not always a word onset. Furthermore, in each sentence, it was required that at least one odd term was not a word onset.

## Speech

Each story was either synthesized as an isochronous sequence of syllables or naturally read by a human speaker.

### Isochronous speech

All syllables were synthesized independently using the Neospeech synthesizer (http://www.neo-speech.com/, the male voice, Liang). The synthesized syllables were 75–354 ms in duration (mean duration 224 ms). All syllables were adjusted to 250 ms by truncation or padding silence at the end, following the procedure in *Ding et al., 2016a*. The last 25 ms of each syllable were smoothed by a cosine window and all syllables were equalized in intensity. In this way, the syllables were presented at a constant rate of 4 Hz (*Figure 1A and B*). In addition, a silence gap lasting 500 ms, that is, the duration of two syllables, was inserted at the position of any punctuation, to facilitate story comprehension.

### Natural speech

The stories were read in a natural manner by a female speaker, who was not aware of the purpose of the study. In natural speech, syllables were not produced at a constant rate and the boundaries between syllables were labeled by professionals (*Figure 1C*). The total duration of speech was 1122 s for the 21 metrical stories and 372 s for the seven nonmetrical stories. A behavioral test showed that most participants did not perceive any difference between the metrical and nonmetrical stories (*Supplementary file 1*).

### Amplitude-modulated speech

Amplitude modulation (AM) was applied to isochronous speech to create a word-rate acoustic rhythm (*Figure 1D*). In σ1-amplified condition, all σ1 syllables were amplified by a factor of 4. In σ2-amplified condition, all σ2 syllables were amplified by a factor of 4. Such 2-Hz AM was clearly per-ceivable but did not affect speech intelligibility, since sound intensity is a weak cue for stress (*Zhong et al., 2001*) and stress does not affect word recognition in Mandarin Chinese (*Duanmu, 2001*). A behavioral test suggested that when listening to amplitude-modulated speech, a larger number of participants perceived σ1-amplified speech as more natural than σ2-amplified speech (*Supplementary file 1*).

## Experimental procedures and tasks

### Behavioral test

A behavioral test was conducted to assess the naturalness of the stimuli. The test was divided into two blocks. In block 1, the participants listened to a metrical story and a nonmetrical story read by a human speaker, which were presented in a pseudorandom order. The stories were randomly selected from the story set. Each story ranged from 53 to 66 s in duration. After listening to each story, the participants were asked to write a sentence to summarize the story and fill out a question-naire. Block 2 was of the same procedure as block 1, except that the metrical and nonmetrical sto-ries were replaced with σ1- and σ2-amplified speech.

In block 1, the first question in the questionnaire asked whether the two types of stories, a metri-cal and a nonmetrical story, showed any noticeable difference regardless of their content. Thirty-one participants (91%) reported no difference perceived, and the other three participants (9%) were asked to elaborate on the differences they detected. Two of them said the metrical story showed larger pitch variations, and the other said they were read with a different tone. Therefore, the vast majority of the participants did not notice any difference between the two types of stories read by a human speaker. A few participants noticed some differences in the intonation pattern but no partici-pants reported the difference in word rhythm.

The second question was to check the naturalness of the stories read. Twenty-four participants (71%) reported that both the two types of stories were naturally read. Three participants (9%) commented that metrical story was not naturally read, and all of them attributed it to intonation. The rest seven participants (20%) thought neither types of stories were naturally read. The reasons reported included (1) exaggerated intonation ($N = 2$); (2) the speed and intonation pattern seemed uniform ($N = 2$); (3) lack of emotion ($N = 2$); and (4) the pitch went up at the end of each sentence ($N = 1$). In sum, most participants thought the stories were naturally read and only two participants (6%) commented on the uniformity of pace.

In block 2, only one question was asked. Participants had to compare the naturalness and accessibility of $\sigma 1$-amplified speech or $\sigma 2$-amplified speech. Fifteen participants (44%) perceived $\sigma 1$-amplified speech as being more natural, two participants (6%) perceived the $\sigma 2$-amplified speech as being more natural, and the rest 17 participants (50%) thought there was no difference in naturalness between the two conditions. In sum, relatively more participants thought the $\sigma 1$-amplified speech was more natural.

## EEG experiment

The study consisted of four EEG experiments. Experiments 1–3 involved 16 participants respectively, and Experiment 4 involved 20 participants. Experiments 1 and 4 both consisted of two blocks with stories presented in a randomized order within each block. In Experiments 2 and 3, all stories were presented in a randomized order in a single block.

### Experiment 1

Synthesized speech was presented in Experiment 1. The experiment was divided into two blocks. In block 1, participants listened to isochronous speech, including seven metrical stories and seven nonmetrical stories. In block 2, the participants listened to amplitude-modulated speech, including 7 $\sigma 1$-amplified stories and 7 $\sigma 2$-amplified stories. All 14 stories presented in block 2 were metrical stories and did not overlap with the stories used in block 1. Participants were asked to keep their eyes closed while listening to the stories. After listening to each story, participants were required to answer three comprehension questions by giving oral responses. An experimenter recorded the responses and then pressed a key to continue the experiment. The next story was presented after an interval randomized between 1 and 2 s (uniform distribution) after the key press. The participants took a break between blocks.

### Experiment 2

Speech stimuli used in Experiment 2 were the same as those used in Experiment 1, while the task was different. The participants were asked to watch a silent movie (The Little Prince) with subtitles and ignored any sound during the experiment. The stories were presented ~5 min after the movie started to make sure that participants were already engaged in the movie watching task. The interval between stories was randomized between 1 and 2 s (uniform distribution). The movie was stopped after all 28 stories were presented. The experiment was followed up with questions on the awareness of stories being presented during the movie watching task, and 87.5% participants ($N = 14$) reported that they did not notice any story.

### Experiment 3

Experiment 3 used the same set of stories as those used in Experiment 1, but the stories were naturally read by a human speaker. The task was of the same design as that in Experiment 1. Participants listened to 21 metrical stories and seven nonmetrical stories. Participants took a break after every 14 stories.

### Experiment 4

Experiment 4 was designed to test whether the results based on amplitude-modulated speech was replicable in a different group of participants. All stories used in Experiment 4 were metrical stories and each story was presented once. In block 1, participants were asked to watch a silent movie (The Little Prince) with subtitles and ignored any sound during the task. Amplitude-modulated speech (5 $\sigma 1$-amplified and 5 $\sigma 2$-amplified stories) were presented ~5 min after the movie started. The interval

between stories was randomized between 1 and 2 s (uniform distribution). Block 1 was followed up with questions on the awareness of stories being presented during the movie watching task, and 15% participants ($N$ = 3) reported that they did not notice any story during the task. Note that the percentage of participants reporting no awareness of the presentation of stories was much lower in Experiment 4 than that in Experiment 2. A potential explanation was that Experiment 4 only presented amplitude-modulated speech and the consistent presence of word-rate acoustic cues facilitated word recognition. In Experiment 2, however, as amplitude-modulated speech was mixed with isochronous speech, the lack of consistent presence of AM cue diminished its effect. After block 1 was finished, the participants took a break.

In block 2, participants listened to amplitude-modulated speech (5 $\sigma$1-amplified stories and 5 $\sigma$2-amplified stories) with their eyes closed. At the end of each story, three comprehension questions were presented, and answers were to be given with oral responses. The experimenter recorded the responses and then pressed a key to continue the experiment. The next story was presented after an interval randomized between 1 and 2 s (uniform distribution) after the key press.

## Data recording and preprocessing

Electroencephalogram (EEG) and electrooculogram (EOG) were recorded using a Biosemi Active-Two system. Sixty-four EEG electrodes were recorded. Two additional electrodes were placed at the left and right temples to record the horizontal EOG (right minus left), and two electrodes were placed above and below the right eye to record the vertical EOG (upper minus lower). Two additional electrodes were placed at the left and right mastoids and their average was used as the reference for EEG (*Ding et al., 2018*). The EEG/EOG recordings were low-pass filtered below 400 Hz and sampled at 2048 Hz.

All preprocessing and analysis in this study were performed using MATLAB (The MathWorks, Natick, MA). The EEG recordings were down-sampled to 128 Hz, referenced to the average of mastoid recordings, and band-pass filtered between 0.8 Hz and 30 Hz using a linear-phase finite impulse response (FIR) filter (6 s Hamming window, $-6$ dB attenuation at the cut-off frequencies). A linear-phase FIR filter causes a constant time delay to the input. The delay equals to $N/2$, where $N$ was the window length of the filter (*Oppenheim et al., 1997*). The delay was compensated by removing the first $N/2$ samples in the filter output. To remove ocular artifacts in EEG, the horizontal and vertical EOG were regressed out using the least-squares method (*Ding et al., 2018*). Occasional large artifacts in EEG/EOG, that is, samples with magnitude >1 mV, were removed from the analysis (*Jin et al., 2018*).

## Data analysis

### Frequency-domain analysis

The EEG responses during the gap between sentences and the responses to the first two syllables of each sentence were removed from analysis to avoid the responses to sound onsets. The EEG responses to the rest of the sentence, that is, from the third syllable to the last syllable of each sentence, were then concatenated.

To further remove potential artifacts, the EEG responses were divided into 7 s trials (a total of 148 trials for Experiments 1–3, and 104 trials for Experiment 4), and we visually inspected all the trials and removed trials with identifiable artifacts. On average, 8.45 ± 3.20% trials were rejected in Experiment 1, 15.20% ± 3.97% trials were rejected in Experiment 2, 10.35% ± 1.53% trials were rejected in Experiment 3, and 12.9 ± 4.46% trials were rejected in Experiment 4.

Then, the response averaged over trials was transformed into the frequency-domain using Discrete Fourier transform (DFT) without any additional smoothing window. Therefore, the frequency resolution of the DFT analysis was 1/7 Hz. The response power, that is, the square of the magnitude of the Fourier coefficients, was grand averaged over EEG electrodes and participants. The phase of the Fourier coefficients were averaged using the circular mean (*Fisher, 1995*). The 2-Hz phase difference between the $\sigma$1- and $\sigma$2-amplified conditions was averaged over participants in each electrode.

## Time-warping analysis

In natural speech used in Experiment 3, syllables were not produced at a constant rate, and therefore the responses to syllables and words were not frequency tagged. However, the neural response to natural speech could be time warped to simulate the response to isochronous speech (*Jin et al., 2018*). In the time-warping analysis, we first extracted the ERP response to each syllable (from 0 to 750 ms), and simulated the response to 4-Hz isochronous speech using the following convolution procedure: $s(t) = \Sigma_j\, h_j(t)*\delta(t - 0.25j)$, where $s(t)$ was the time-warped response, $\delta(t)$ was the Dirac delta function, and $h_j(t)$ was the ERP evoked by the $j^{th}$ syllable. The word index $j$ ranged from one to the total number of syllables in a story.

In the time-warping procedure, it was assumed that the syllable response was time-locked to the syllable onsets and the word response was time-locked to word onsets. The frequency-domain analysis was subsequently applied to the time-warped response, following the same procedure as adopted in the analysis of the response to isochronous speech.

## Time-domain analysis

Time-domain analysis was only applied to the responses to disyllabic words, and the responses to monosyllabic words were not analyzed. The ERP responses to the first and second syllables of each disyllabic word were separately extracted and averaged across all disyllabic words. The ERP response to each syllable was baseline correlated by subtracting the mean response in a 100 ms window before the syllable onset.

## Statistical test

In the frequency-domain analysis, statistical tests were performed using bias-corrected and accelerated bootstrap (*Efron and Tibshirani, 1994*). In the bootstrap procedure, data of all participants were resampled with replacement 10,000 times. To test the significance of the 2-Hz and 4-Hz peaks in the response spectrum (*Figures 2A–E*, *3B and D*, and *4A and B*), the response amplitude at the peak frequency was compared with the mean power of the neighboring four frequency bins (two bin on each side, one-sided comparison). If the response power at 2 Hz or 4 Hz was stronger than the mean power of the neighboring bins $N$ times in the resampled data, the significance level could be calculated as $(N + 1)/10,001$ (*Jin et al., 2018*).

When comparing the response power between conditions, the response power was always subtracted by the power averaged over four neighboring frequency bins (two on each side) to reduce the influence of background neural activity. A two-sided test was used to test the power difference between conditions within an experiment (solid black lines in *Figure 3A and B* and *Figures 4E* and *5C*; topography in *Figures 4F* and *5D*). If the response power was greater in one condition $N$ times in the resampled data, the significance level could be calculated as $(2N +1)/10,001$. As to the power difference between experiments (dotted red lines in *Figures 3A and B* and *Figure 4E*), the significance level was $v$ if the sample mean in one experiment exceeded the 100 $v/2$ percentile (or fell below the $v/2$ percentile) of the distribution of the sample mean in the other experiment (*Ding et al., 2018*).

To test the phase difference between conditions, the $V\%$ confidence interval of the phase difference was measured by the smallest angle that could cover $V\%$ of the 10,000 resampled phase difference (*Jin et al., 2018*). In the inter-participant phase coherence test (*Figure 3—figure supplement 1*), 10,000 phase coherence values were generated based on the null distribution, that is, a uniform distribution. If the actual phase coherence was smaller than $N$ of the 10,000 phase coherence values generated based on the null distribution, its significance level was $(N + 1)/10,001$ (*Ding et al., 2018*).

In the time-domain analysis (*Figure 6*), the significance of ERP difference between conditions was determined by means of the cluster-based permutation test (*Maris and Oostenveld, 2007*). The test was performed with the following steps given below: (1) The ERP for each participant in two conditions were pooled into the same set. (2) The set was randomly partitioned into two equally sized subsets. (3) At each time point, the responses were compared between the two subsets using a paired t-test. (4) The significantly different data points in the responses were clustered based on temporal adjacency. (5) The cluster-level statistics were calculated by taking the sum over the

t-values within each cluster. (6) Steps 2–5 were repeated 2000 times. The p-value was estimated as the proportion of partitions that resulted in a higher cluster-level statistic than the actual two conditions.

When multiple comparisons were performed, the p-value was adjusted using the false discovery rate (FDR) correction (*Benjamini and Hochberg, 1995*).

### Post-hoc effect size calculation

On top of showing the 2-Hz response power from individual participants and individual electrodes in *Figure 2—figure supplement 1*, an effect size analysis was applied to validate that the sample size was appropriate to observe the 2-Hz response. To simplify the analysis, we calculated the effect size based on a paired t-test to compare the power at 2 Hz and the power averaged over four neighboring frequencies. Since the response power was not subject to a normal distribution, such a t-test had lower power than, for example, the bootstrap test. However, based on the t-test, the 2-Hz response remained significantly stronger than the mean response averaged over neighboring frequency bins in all conditions shown in *Supplementary file 2*. The effect size of the t-test was calculated using the G*Power software (version 3.1) (*Faul et al., 2007*). We calculated $d$ and power based on the mean and standard deviation of the 2-Hz response (reported in *Supplementary file 2*). The power was above 0.8 for all conditions, suggesting that the sample size was big enough even for the more conservative t-test.

## Acknowledgements

We thank Dr. Virginie van Wassenhove and three anonymous reviewers for their constructive comments. We thank Dr. Xunyi Pan, Dr. Lang Qin, Jiajie Zou, and Yuhan Lu for thoughtful comments on previous versions of the manuscript. The research was supported by National Natural Science Foundation of China 31771248 (ND), Major Scientific Research Project of Zhejiang Lab 2019KB0AC02 (ND), National Key R and D Program of China 2019YFC0118200 (ND), Zhejiang Provincial Natural Science Foundation of China LGF19H090020 (CL), and Fundamental Research Funds for the Central Universities 2020FZZX001-05 (ND).

## Additional information

### Funding

| Funder | Grant reference number | Author |
|---|---|---|
| National Natural Science Foundation of China | 31771248 | Nai Ding |
| MajorScientific Research Project of Zhejiang Lab | 2019KB0AC02 | Nai Ding |
| Zhejiang Provincial Natural Science Foundation | LGF19H090020 | Cheng Luo |
| Fundamental Research Funds for the Central Universities | 2020FZZX001-05 | Nai Ding |
| National Key R&D Program Of China | 2019YFC0118200 | Nai Ding |

The funders had no role in study design, data collection and interpretation, or the decision to submit the work for publication.

### Author contributions

Cheng Luo, Conceptualization, Data curation, Software, Formal analysis, Investigation, Visualization, Methodology, Writing - original draft; Nai Ding, Conceptualization, Formal analysis, Supervision, Validation, Visualization, Methodology, Project administration, Writing - review and editing

## Author ORCIDs

Nai Ding https://orcid.org/0000-0003-3428-2723

## Ethics

Human subjects: The experimental procedures were approved by the Research Ethics Committee of the College of Medicine, Zhejiang University (2019-047). All participants provided written informed consent prior to the experiment and were paid.

## Decision letter and Author response

Decision letter https://doi.org/10.7554/eLife.60433.sa1
Author response https://doi.org/10.7554/eLife.60433.sa2

## Additional files

### Supplementary files

• Source code 1. The MATLAB code to process data in Experiments 1–3 and plot the results as displayed in *Figure 2*.

• Source code 2. The MATLAB code to process data in Experiments 1–3 and plot the results as displayed in *Figure 3*.

• Source code 3. The MATLAB code to process data in Experiments 1–3 and plot the results as displayed in *Figure 4*.

• Source code 4. The MATLAB code to process data in Experiment 4 and plot the results as displayed in *Figure 5*.

• Source code 5. The MATLAB code to process data in Experiments 1–3 and plot the results as displayed in *Figure 6*.

• Source code 6. MATLAB functions used in other source codes.

• Supplementary file 1. Assessment of the stimulus.

• Supplementary file 2. Post-hoc effect size calculation.

• Supplementary file 3. Samples of the stimulus including isochronous speech, natural speech, $\sigma 1$-amplified speech, and $\sigma 2$-amplified speech.

• Transparent reporting form

### Data availability

The EEG data and analysis code (in MatLab) were uploaded as source data files.

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
