## [Decision Letter]

Thank you for submitting your article "Delta-band Cortical Tracking of Acoustic and Linguistic Features in Natural Spoken Narratives" for consideration by *eLife*. Your article has been reviewed by three peer reviewers, one of whom is a member of our Board of Reviewing Editors, and the evaluation has been overseen by Andrew King as the Senior Editor. The reviewers have opted to remain anonymous.

The reviewers have discussed the reviews with one another and the Reviewing Editor has drafted this decision to help you prepare a revised submission.

Summary:

The acoustics of human speech convey complex linguistic hierarchical structures. While several studies have demonstrated that cortical activity shows speech-following responses, the extent to which these responses track acoustic features or linguistic units remains highly debated. Delta-band activity has been reported to track prosody, syntactic phrasing as well as temporal predictions. In this new study comprised of three EEG experiments, Luo and colleagues tested the cortical responses to the presentation of spoken stories which could be metrically organized or not, and read via synthesized isochronous or delta (~2Hz) amplitude modulated isochronous speech, or by a natural reader. The authors argue that delta synchronizes to multisyllabic compound words, favoring a word-based rather than a syllabic-based speech tracking response.

1) The Materials and methods section and the figures need to be clearly documented and annotated. Reviewer 1 raises some concerns about the loose use of "phase" and some of the methods that are not clearly incorporated in the Results to help the more general audience follow the subtleties of the analyses. I also leave the full set of reviews to guide such revision – as well as edit the English when needed.

2) Tempering the claim that delta uniquely tracks word rhythm.

All reviewers provide detailed comments pointing out to additional literature that could be considered and which can feed the discussion regarding syntactic vs. semantic vs. prosodic dissociations…

Clarifying the specificities reported in the literature should also help the authors address serious comments raised by reviewer 1 and 3, who are particularly hesitant regarding the strong claim on word-specificity of the delta response – including the lack of changes in the phase of delta oscillations. Reviewer 2 also stresses this point. The legitimacy of calling the low-frequency response delta is questioned as opposed to a cautionary possible mixture of network activity as suggested for instance by reviewer 3.

All three reviewers further question the description of delta phase considering the quantifications (e.g. well-known effect of the reference) and the observable topographical changes in Figure 3.

As a result of these major considerations, the authors should temper the strong claims that are currently being put forward.

Revisions expected in follow-up work:

3) While the study provides an interesting means for parametric manipulation of amplitude, reviewer 2 raised an interesting point regarding the effect of prosody in that frequency range. As expressed by reviewer 2: "no only the amplitude of pitch changes is critical to the perception of prosodic events, but also the timing and rise/fall slopes. All this is not linearly affected by increasing the volume at 2 Hz. To show that delta-band oscillations are not affected by stimulus prosody would require a more systematic dedication to the different linguistic dimension of prosody, amplitude being only one."

Reviewer #1:

In a series of three EEG experiments, Luo and colleagues tested whether the low-frequency following response to auditory speech was elicited by linguistic or acoustic features. The authors modulated the task (story comprehension vs. passive video watching) and the acoustic structure of the speech signals using (non-)metrical, isochronous, and 2Hz amplitude modulated speech, or natural speech. The authors mainly used spectral quantification methods of the following responses and some phase analyses. The authors summarize their findings as illustrating that the delta band synchronizes to multisyllabic words, favoring a word-based following response as compared to a syllabic-based speech tracking response.

My main concerns about the study stem from the methods and a difficulty to straightforwardly understand the procedure.

For instance:

– Can the authors clarify the choice of band-pass filtering and the compensation that was applied as briefly stated "and the delay caused by the filter was compensated."

– How did the authors insure that concatenated responses did not introduce low-frequency noise in their analysis?

– Subsection “Frequency domain analysis”: can the authors report the number of participants excluded by the analysis?

I also have substantial difficulties regarding the following methodological aspects:

– Bootstrap analyses artificially increase the number of samples while maintaining constant the variance of the samples. How did the authors take this into account in their final statistical comparisons?

– Figure 1D and predictions on the (absolute? relative? Instantaneous?) phase-locked activity to word onset and syllable need to be better and more precisely explained. For instance, there is no a priori reason to believe that 0° and 180° owe to be the actual expected values as clearly illustrated in Figure 3?

– While a time-warping analysis is being performed for data collected in Experiment 3, no mention of this approach is being explained in more details (as to possible interpretational limits) in the actual Results section.

I am here particularly concerned about the phase analysis which is not being sufficiently reported in details to clearly understand the results and the claims of the authors.

In line with this, I highlight two additional issues:

– Subsection “Response phase at 2 Hz”, Figure 4: the authors argue that attention may explain the lack of phase differences. Additionally, the authors seem to assume in their working hypothesis that the phase response should be comparable across participants.

In a recent study (Kösem et al., 2016), no evidence for phase-related linguistic parsing in low-frequency activity was found when participants actively or passively listened to a stationary acoustic speech sequence whose perceptual interpretation could alternate. Are these findings compatible with those illustrated in Figure 4? Furthermore, some important inter-individual differences were reported, which could severely impact the primary assumption of phase-consistency across individuals.

– Have the authors considered the possibility that the choice of EEG reference may largely affect the reported mean instantaneous phase responses and if so, could they control for it?

Reviewer #2:

The authors report an n = 48 EEG study aiming at dissociating the previously proposed roles of delta-band oscillations in the processing of speech acoustics and symbolic linguistic information. The authors report that delta-band activity is related more strongly to linguistic rather than acoustic information.

This is a welcome study, given that by now, the literature contains claims of delta-band oscillations being involved in (1) prosody tracking, (2) generation of linguistic structures / syntactic phrasing, (3) processing of linguistic information content, (4) temporal prediction in- and outside of the linguistic domain.

The study is well-written and I am optimistic that it will provide a good contribution to the existing literature.

I find the framing and discussion a bit gappy and I missed multiple lines of related research. For instance, the literature on the relationship between delta-band oscillations and prosody is not mentioned in the Introduction; likewise, the literature on temporal prediction versus linguistic prediction is not discussed well. In addition, I also think that this study is *not* as definitive as the authors claim, resulting from the choice of unnatural (i.e., isochronous) stimuli that, strangely enough, the authors claim to be natural.

Below, I given some comments for the authors to further improve their interesting work.

1) False claim of studying natural speech: The authors artificially induce unnatural stimulus rhythms as a means to frequency-tag their stimuli and perform frequency-domain analysis. While this is an interesting approach, this results in stimuli that do not have ecological validity. Using unnatural stimuli with artificial periodicity to propose periodic electrophysiological processing of words is not scientifically sound. In natural speech, words do *not* occur periodically. Why assume then that they are processed by a periodic oscillatory mechanism? While the frequency-tagging approach inherently provides good SNR, it cannot claim ecological validity for anything above phonemes or syllables, as for these, there are no linguistic corpus studies out there that show periodicity of the constructs to start with. Specific comment:

Subsection “Neural tracking of words in natural spoken narratives”: "natural" The authors artificially make natural speech unnatural by making it isochronous. It is bizarre to still call the resulting stimulus "natural". Isochronous word sequences are certainly not natural speech. Hence, the authors can also not claim that they demonstrate cortical tracking of natural speech. To show this, they would have to show that there is speech-brain synchronicity for the non-isochronous condition. Certainly, to do so, the authors cannot use their standard spectral-peak analysis, but would have to show coherence or phase-locking against a different baseline (e.g., scrambled, autocorrelation-preserving speech). I strongly encourage the authors to do this analysis instead of pretending that isochronous word sequences are *natural*. The more general problem with this is that it adds to literature claiming that oscillations track words in human speech processing, which cannot possibly be true in this simplistic version: words are simply too variable in duration and timing.

2) Depth of framing and discussion: (a) Introduction is not specific enough to speech prosody versus the different types of results on linguistic information (e.g., syntactic structure / phrasing, syntactic and lexical entropy), including reference to prior work; also, relevant references that suggested delta-band oscillations to serve both bottom-up (i.e., prosody) and top-down (i.e., syntax, prediction) functionalities are not discussed in the Introduction; (b) interpretation of the results in terms of the different proposed functionalities of the delta-band lacks depth and differentiation.

Introduction: What is missing here is mentioning that delta-band oscillations have also been implied in prosody (e.g., Bourguignon et al., 2013). Relatedly, the idea that delta-band oscillations are related both to slow-fluctuating linguistic representations (e.g., phrases; Meyer and Gumbert, 2018; Meyer et al., 2016; Weissbart et al., 2019) and prosody has been put forward various times (e.g., Ghitza, 2020; Meyer et al., 2016). I think that mentioning this would help the framing of the current manuscript: (1) delta = symbolic linguistic units, (2) delta = prosody, (3) current study: Is delta reflecting symbolic processing or prosody?

Subsection “Time course of ERP response to words”: As to prediction and the delta-band, please consider discussing Weissbart et al., 2019, Meyer and Gumbert, 2018, Breska and Deouell, 2017, Lakatos et al., 2013, Roehm et al., 2009; in particular the relationship between linguistic predictions and the delta-band should be discussed shortly here. In general, there is need in the field to characterize the relationship between temporal prediction, prosody, linguistic units, and the delta-band. Just noting all this would be fine here.

"semantic processing" Why semantics? See my above comment on the relationship between the delta-band and temporal predictions (e.g., Roehm et al., 2009; Stefanics et al., 2010). In addition, please consider the recent reconceptualisation of the N400-P600 complex in terms of predictive coding (e.g., Kuperberg et al., 2019).

3) EEG preprocessing / report thereof does not accord to standards in the field. Apparently, the EEG data have not been cleaned from artifacts except for calculating the residual between the EEG and the EOG. Hence, I cannot assume that the data are clean enough to allow for statistical analysis. Specific point:

Materials and methods: The EEG-cleaning section is very sparse. With standard EEG and standard cleaning, rejection rates for adult EEG are in the other of 25-35 %. It am highly doubtful that these EEG data have been cleaned in an appropriate way. There is no mentioning of artifact removal, amount of removed data segments, time- and frequency-domain aspects that were considered for artifact definition. What toolboxes were used, what algorithms?

4) Terminology: Like many people in the field, the authors speak of “tracking” of linguistic information. To be clear: While it is fine that the field understands the problems of the term “entrainment”, now opting for “tracking”, it does not make much sense to use the combination of “tracking” and “linguistic units”. This is because linguistic units are not in the stimulus; instead they are only present in the brain as such. They cannot be tracked; instead, they need to be “associated”, “inferred”, or even “generated”. If one were to assume that the brain “tracks” linguistic units, this would mean that the brain tracks itself. Specific comments:

"cortical activity tracking of higher-level linguistic units" Linguistic units, such as "words and phrases" cannot be tracked, as they are not present in the stimulus. They are purely symbolic and they must be inferred. They cannot be sensed. Instead, they are present only in the mind and electrophysiology of the listener. Linguistic units are activated in memory or generated in some form. They are internal representations or states. If one were to say that the brain tracks these in some way, one would say that the brain tracks its own internal representations / states. Try and rephrase, please.

"linguistic information" cannot be extracted from acoustic features. Instead, linguistic information (e.g., distinctive features in phonology, semantic meaning, part-of-speech labels) must be associated with a given acoustic stimulus by an associative / inferential cognitive process. When we hear some acoustic stimulus, we associate with something linguistic. Just as in my above comment on the loose use of the term “tracking”, I suggest the authors to change this wording as well. A suggestion for a better phrasing would be "Linguistic information is retrieved / constructed / generated when specific acoustic information is encountered / recognized…". Or "Speech comprehension involves the active inference of linguistic structure from speech acoustics".

Discussion: "tracks" see my above comment.

5) Potential differences between the σ1 and 2 conditions: There are apparent differences in the number of sensors that show sensitivity to the amplitude modulation. This is not captured in the current statistical analysis. Still, given these drastic differences, the claim that acoustics of speech prosody do not affect delta-band amplitude is certainly much too strong and may just result from a lack of analysis (i.e., overall phase / ERP rather than sensor-by-sensor analysis). See specific comment next:

"or the phase of the 2-Hz amplitude modulation" I disagree with excitement and a happy suggestion that the authors must consider. Look, the plots (Figure 3, σ1/2) may not show differences in the phase angle. Yet, what they may point to is a dimension of the data that the authors have not assessed yet: topographical spread. From simply eyeballing it is clear that the number of significant electrodes differs *drastically* between the σ1 and σ2 conditions. This pattern is interesting and should be examined more closely. Is word-initial stress more natural in Chinese? If so, this could be interpreted in terms of some influence of native prosodic structure: Amplitude modulation does appear affect the amount / breadth of EEG signal (i.e. number of sensors), but only if it conforms to the native stress pattern? Not only words / phrases are part of a speakers linguistic knowledge, but also the stress pattern of their language. I urge the authors to look into this more closely.

"The response" see my above comment on the differences in the number significant sensors in the σ1 and 2 conditions.

Reviewer #3:

This manuscript presents research aimed at examining how cortical activity tracks with the acoustic and linguistic features of speech. In particular, the authors are interested in how rhythms in cortical data track with the timing of the acoustic envelope of speech and how that might reflect and, indeed, dissociate, from the tracking of the linguistic content. The authors collect EEG from subjects as they listen to speech that is presented in a number of different ways. This includes conditions where: the speech is rhythmic and the first syllable of every word appears at regular intervals; the speech is rhythmic but the first syllable of every word does not appear at regular intervals; the speech appears with a more natural rhythm with some approximate regularity to the timing of syllables (that allows for an analysis based on time warping); and some rhythmic speech that is amplitude modulated to either emphasize the first or second syllables. And they also present these stimuli when subjects are tasked with attending to the speech and when subjects are engaged in watching a silent video. The authors analyze the EEG from these different conditions in the frequency domain (using time-warping where needed), in terms of the phase of the rhythmic EEG responses; and in the time domain. Ultimately, they conclude that, when subjects are attending to the speech, the EEG primarily tracks the linguistic content of speech. And, when subjects are not attending to the speech, the EEG primarily tracks the acoustics.

I enjoyed reading this manuscript, which tackles an interesting and topical question. And I would like to commend the authors on a very nice set of experiments – it's a lovely design.

However, I also had a number of fairly substantial concerns – primarily about the conclusions being drawn from the data.

1) My main concern with the manuscript is that it seems – a priori – very committed to discussing the results in terms of tracking by activity in a specific (i.e., delta) cortical band. I understand the reason to want to do this – there is a lot of literature on speech and language processing that argues for the central role of cortical oscillations in tracking units of speech/language with different timescales. However, in the present work, I think this framework is causing confusion. To argue that delta tracking is a unitary thing that might wax and wane like some kind of push/pull mechanism to track acoustics sometimes and linguistic features other times seems unlikely to be true and likely to cause confusion in the interpretation of a nice set of results. My personal bias here – and I think it could be added for balance to the manuscript – is that speech processing is carried out by a hierarchically organized, interconnected network with earlier stages/lower levels operating on acoustic features and later stages/higher levels operating on linguistic features (with lots of interaction between the stages/levels). In that framing, the data here would be interpreted as containing separable simultaneous evoked contributions from processing at different hierarchical levels that are time locked to the relevant features in the stimulus. Of course, because of the nature of the experiments here, these contributions are all coming at 2 Hz. But to then jump to saying a single "delta oscillation" is preferentially tracking different features of the speech given different tasks, seems to me to be very unlikely. Indeed, the authors seem sensitive to the idea of different evoked contributions as they include some discussion of the N400 later in their manuscript. But then they still hew to the idea that what they are seeing is a shift in what "delta-band" activity is doing. I am afraid I don't buy it. I think if you want to make this case – it would only be fair to discuss the alternatives and more clearly argue why you think this is the best way to interpret the data.

2) Following my previous point, I have to admit that I am struggling to figure out how, precisely, you are determining that "cortical activity primarily tracks the word rhythm during speech comprehension". There is no question that the cortical activity is tracking the word rhythm. But how are you arguing that it is primarily word rhythm. I am guessing it has something to do with Figure 3? But I am afraid I am not convinced of some of the claims you are making in Figure 3. You seem to want to embrace the idea that there is no difference in phase between the σ1-amplified and σ2-amplified conditions and that, therefore, the signal is primarily tracking the word rhythm. But I think there is an important flaw in this analysis – that stems from my concern in point number 1 above. In particular, I think it is pretty clear that there are differences in phase at some more frontal channel locations, and no difference in more posterior channels (Figure 3A, rightmost panel). So, going back to my issue in point number 1, I think it is very likely that word rhythm is being tracked posteriorly (maybe driven by N400 like evoked activity) – and that phase is similar (obviously). But it also seems likely that the 2 Hz rhythm over frontal channels (which likely reflect evoked activity based on acoustic features) are at a different phase. Now, cleanly disentangling these things is difficult because they are both at 2Hz and will contaminate each other. And I think when subjects are watching the silent video, the posterior (N400 like) contribution disappears and all you have left is the acoustic one. So, again, I think two separable things are going on here – not just one delta rhythm that selectively tracks different features.

3) I'm afraid I must take issue with the statements that "Nevertheless, to dissociate linguistic units with the related acoustic cues, the studies showing linguistic tracking responses mostly employ well-controlled synthesized speech that is presented as an isochronous sequence of syllables… Therefore, it remains unclear whether neural activity can track linguistic units in natural speech, which is semantically coherent but not periodic, containing both acoustic and linguistic information in the delta band." I know the (very nice) studies you are talking about. But you have just also cited several studies that try to disentangle linguistic and acoustic features using more natural speech (e.g., Brodbeck; Broderick). So I think your statement is just too strong.

4) I was confused with the presentation of the data in Figure 2F and G. Why are some bars plotted up and others down? They are all positive measures of power, and they would be much easier to compare if they were all pointed in the same direction. Unless I am missing the value of plotting them this way?

---

## [Author Response]

Revisions for this paper:1) The Materials and methods section and the figures need to be clearly documented and annotated. Reviewer 1 raises some concerns about the loose use of "phase" and some of the methods that are not clearly incorporated in the Results to help the more general audience follow the subtleties of the analyses. I also leave the full set of reviews to guide such revision – as well as edit the English when needed.

Based on the very constructive comments from the reviewers, we have clarified our analysis procedures in the revised manuscript. The response phase in the manuscript refers to the phase of the complex-valued Fourier coefficients, which we have now clearly defined. Furthermore, we have also added a new analysis that is easier to interpret and moved detailed results of phase analysis to Figure 3—figure supplement 1 (please see the reply to the next question for details).

2) Tempering the claim that delta uniquely tracks word rhythm.All reviewers provide detailed comments pointing out to additional literature that could be considered and which can feed the discussion regarding syntactic vs. semantic vs. prosodic dissociations…

We have added the suggested references to the manuscript and expanded the Introduction and Discussion. Please see our point-to-point reply for details.

Clarifying the specificities reported in the literature should also help the authors address serious comments raised by reviewer 1 and 3, who are particularly hesitant regarding the strong claim on word-specificity of the delta response – including the lack of changes in the phase of delta oscillations. Reviewer 2 also stresses this point. The legitimacy of calling the low-frequency response delta is questioned as opposed to a cautionary possible mixture of network activity as suggested for instance by reviewer 3.

In the previous manuscript, we referred to the 2-Hz neural response as a “delta-band” neural response to distinguish it from the 4-Hz response to speech envelope which was usually referred to as a “theta-band” envelope-tracking response. We now realized that this terminology was not appropriate since it could mislead the readers to believe that the 2-Hz response must relate to spontaneous delta oscillations. To avoid this potential confusion, we have now used amplitude modulation (AM) and speech envelope to refer to the 2-Hz and 4-Hz acoustic rhythms respectively in the amplitude-modulated speech.

All three reviewers further question the description of delta phase considering the quantifications (e.g. well-known effect of the reference) and the observable topographical changes in Figure 3.As a result of these major considerations, the authors should temper the strong claims that are currently being put forward.

The original conclusion that “delta-band activity primarily tracks the word rhythm” was made on the basis of response phase analysis, which showed that the response phase was barely influenced by the 2-Hz amplitude modulation (AM). The three reviewers, however, all raised concerns about the conclusion and/or the relevant analysis procedure. Following the constructive comments from the reviewers, we have now collected more data and performed new analyses to further investigate how acoustic and linguistic information is jointly encoded in cortical activity. We have updated our conclusions on the grounds of the new results, which are summarized in the following.

Phase difference analysis

Reviewers 2 and 3 both pointed out that the response phase was affected by the 2-Hz AM even during attentive story comprehension. To further investigate this important observation, we have now calculated the phase difference between σ1- and σ2-amplified conditions for each participant and each EEG electrode, and tested whether the phase difference was significantly different from 0º in any electrode. The results indeed revealed a significant phase difference between conditions in some EEG electrodes, even during the story comprehension task (Author response image 1, B), confirming the reviewers’ observation.

**Author response image 1. sa2fig1:** Topographical distribution of the 2-Hz phase difference between the σ1- and σ2-amplified conditions. The phase difference is calculated for each participant and each electrode, and then averaged over participants. The black dots indicate electrodes showing a significant phase difference between the σ1- and σ2-amplified conditions (P < 0.05, bootstrap, FDR corrected). (AB) The 2-Hz phase difference in the original experiment. (CD) The 2-Hz phase difference pooled over the original experiment and the replication experiment.

A replication experiment

To further validate the results of phase difference analysis, we have now replicated the experiment with another group of participants. In this replication experiment, participants only listened to amplitude-modulated speech and they participated in the video watching task before the story comprehension task. The replication experiment confirmed our original observation, finding larger 2-Hz response phase difference between the σ1- and σ2-amplified conditions in the video watching task than in the story comprehension task. Nevertheless, the new observation illustrated in Author response image 1 was not replicated. After the data was pooled over the original experiment and the replication experiment, no EEG electrode showed any significant phase difference between conditions during the story comprehension task (Author response image 1).

Therefore, the 2-Hz AM reliably modulated the response phase during passive listening (Author response image 1, D), but not during active listening (Author response image 1, C). Although the 2-Hz AM did not significantly modulate the response phase during active listening, the following new analysis showed that this null result was potentially attributable to the low statistical power of the response phase analysis.

Time-domain separation of AM and word responses

For amplitude modulated speech, the neural response synchronous to the 2Hz AM (referred to as the AM response) and the response synchronous to word onsets (referred to as the word response) can be dissociated based on response timing (Figure 1D). In the previous manuscript, we mainly employed the phase analysis to characterize response timing (previous Figure 3). Reviewer 1, however, raised the concern that the previous analysis was built on the assumption that the response phase was consistent across participants, which was neither a necessary nor a well-supported assumption. Therefore, we have now employed a different method to extract the AM and word responses based on response timing. This analysis was briefly reported in Supplementary Figure 2 of the previous manuscript. We have now expanded it and moved it to the main text.

In this new analysis, AM and word responses were extracted by averaging the response over the σ1- and σ2-amplified conditions in different ways, which was illustrated in Figure 4A and C. The spectra of the AM and word responses were calculated in the same way as the spectra were calculated for, e.g., the response to isochronous speech. In this new analysis, the power spectrum was calculated for each participant and then averaged. Therefore it did not assume phase consistency across participants. More importantly, this new method could estimate the strength of the AM and word responses, while the previous phase analysis could only prove the existence of the AM response.

The new analysis clearly revealed that the AM response was statistically significant during both attentive and passive listening, suggesting that the cortical response contained a component that reflected low-level auditory encoding. The new analysis confirmed that the word response was stronger than the AM response. Nevertheless, it also showed that the word response was also statistically significant during passive listening for amplitude-modulated speech, but not for isochronous speech, suggesting that the AM cue could facilitate word processing during passive listening. Additionally, the AM response was not significantly modulated by attention, while the word response was. The AM and word responses also showed distinguishable spatial distribution, suggesting different neural sources. These key findings were all replicated in the replication experiment (Figure 5), demonstrating that the new spectral analysis method was robust.

Conclusions

With findings from the new analysis, we have now updated the conclusions in the Abstract as follows.

“Our results indicate that an amplitude modulation (AM) cue for word rhythm enhances the word-level response, but the effect is only observed during passive listening. […] These results suggest that bottom-up acoustic cues and top-down linguistic knowledge separately contribute to cortical encoding of linguistic units in spoken narratives.”

Revisions expected in follow-up work:3) While the study provides an interesting means for parametric manipulation of amplitude, reviewer 2 raised an interesting point regarding the effect of prosody in that frequency range. As expressed by reviewer 2: "no only the amplitude of pitch changes is critical to the perception of prosodic events, but also the timing and rise/fall slopes. All this is not linearly affected by increasing the volume at 2 Hz. To show that delta-band oscillations are not affected by stimulus prosody would require a more systematic dedication to the different linguistic dimension of prosody, amplitude being only one."

Reviewer 2 raised a very important point, which was not discussed in the previous manuscript. We certainly agree that the 2-Hz word response in the current study could reflect prosodic processing. Furthermore, we would like to distinguish the perceived prosody and the acoustic cues for prosody. Previous literature has shown that high-level linguistic information can modulate prosodic and even auditory processing (e.g., Buxó-Lugo and Watson, 2016; Garrett et al., 1966). Therefore, even when the prosodic cues in speech are removed, e.g., in the isochronous speech sequence, listeners may still mentally recover the prosodic structure. In fact, the experiment design is not to rule out the possibility that prosody is related to low-frequency neural activity. This important point was not discussed in the previous manuscript and caused confusion.

Furthermore, as the reviewer pointed out, the amplitude envelope is just one kind of prosodic cues. Furthermore, in most languages, the amplitude envelope contributes less to prosodic processing than other cues, e.g., the pitch contour, rise/fall slopes, and timing. Therefore, the purpose of manipulating the speech envelope is not to modulate prosodic processing. Instead, we manipulate the amplitude envelope since it is one of the acoustic features that can most effectively drive cortical responses. Even amplitude modulated noise, which has no pitch contour or linguistic content, can strongly drive an envelope-tracking response. Therefore, we would like to test how auditory envelope tracking and linguistic processing, which includes prosodic processing, separately contribute to speech synchronous neural activity.

We have now added a new section of discussion about prosodic processing (see point-to-point reply for details) and revised the Discussion to explain why we manipulated the speech envelope.

Reviewer #1:[…] My main concerns about the study stem from the methods and a difficulty to straightforwardly understand the procedure.

Thank you for pointing out these issues. We have thoroughly modified the Materials and methods and Results, and we believe there is now enough information to understand the procedure. Furthermore, we will make the data and analysis code publicly available after the manuscript is accepted.

For instance:– Can the authors clarify the choice of band-pass filtering and the compensation that was applied as briefly stated "and the delay caused by the filter was compensated."

We have now clarified the filter we used and how the delay was compensated.

“The EEG recordings were down-sampled to 128 Hz, referenced to the average of mastoid recordings, and band-pass filtered between 0.8 Hz and 30 Hz using a linear-phase finite impulse response (FIR) filter (6 s Hamming window, -6 dB attenuation at the cut-off frequencies). […] The delay was compensated by removing the first N/2 samples in the filter output…”

– How did the authors insure that concatenated responses did not introduce low-frequency noise in their analysis?

The EEG responses to different sentences were concatenated in the frequency-domain analysis. Duration of the response to each sentence ranged from 1.5 to 7.5 s (mean duration: 2.5 s). Therefore, concatenation of the responses could only generate low-frequency noises below 1 Hz, which could barely interfere with the 2-Hz and 4-Hz responses that we analyzed.

To confirm that the 2-Hz and 4-Hz responses were not caused by data concatenation, we have now also analyzed the EEG response averaged over sentences. Considering the variation in sentence duration, this analysis was restricted to sentences that consisted of at least 10 syllables. Moreover, only the responses to the first 10 syllables in these sentences were averaged. In a procedure similar to that applied in the original spectral analysis, the responses during the first 0.5 s were removed to avoid the onset response and the remaining 2-second response was transformed into the frequency domain using the DFT. The results was consistent with the results obtained from the original analysis and were illustrated in Author response image 2. The advantage of this new analysis was that it did not involve data concatenation, while the disadvantage was that it threw away a lot of data to ensure equal duration in the response to each sentence.

**Author response image 2. sa2fig2:** Spectrum of the EEG response averaged over sentences. This analysis is restricted to sentences that had at least 10 syllables, and only the response to the first 10 syllables is analyzed. The response during the first two syllables is removed to avoid the onset response and the rest 2 seconds of response is averaged over sentences. The averaged response is transformed into the frequency-domain using the DFT. Response spectrum averaged over participants and EEG electrodes. The shaded area indicates 1 standard error of the mean (SEM) across participants. Stars indicate significantly higher power at 2 Hz or 4 Hz than the power averaged over 4 neighboring frequency bins (2 on each side). The color of the star is the same as the color of the spectrum **P < 0.01(bootstrap, FDR corrected).

– Subsection “Frequency domain analysis”: can the authors report the number of participants excluded by the analysis?

In the previous phase analysis, for each EEG electrode, participants not showing significant inter-trial phase coherence (P > 0.1) were excluded. Therefore, a different number of participants were excluded for each electrode. To simplify the procedure, we have now kept all participants in the analysis and the updated results were shown in Figure 3—figure supplement 1.

I also have substantial difficulties regarding the following methodological aspects:– Bootstrap analyses artificially increase the number of samples while maintaining constant the variance of the samples. How did the authors take this into account in their final statistical comparisons?

In the bootstrap procedure, we estimated the distribution of the sample mean by resampling the data. The p-value was determined based on the null distribution. We used a standard bootstrap procedure and its details were specified in the reference (Efron and Tibshirani, 1994), and the code was publicly available (MATLAB scripts: bootstrap_for_vector.m, https://staticcontent.springer.com/esm/art%3A10.1038%2Fs41467-018-07773y/MediaObjects/41467_2018_7773_MOESM4_ESM.zip). The bootstrap procedure is well established and has been used in our previous studies (e.g., Jin et al., 2018) and in other studies (e.g., Bagherzadeh et al., 2020, and Norman-Haignere et al., 2019).

Bagherzadeh, Y., Baldauf, D., Pantazis, D., Desimone, R., 2020. Alpha synchrony and the neurofeedback control of spatial attention. Neuron 105, 577-587. e575.

Norman-Haignere, S.V., Kanwisher, N., McDermott, J.H., Conway, B.R., 2019. Divergence in the functional organization of human and macaque auditory cortex revealed by fMRI responses to harmonic tones. Nature Neuroscience 22, 1057-1060.

– Figure 1D and predictions on the (absolute? relative? Instantaneous?) phase-locked activity to word onset and syllable need to be better and more precisely explained. For instance, there is no a priori reason to believe that 0° and 180° owe to be the actual expected values as clearly illustrated in Figure 3?

Thank you for pointing out this issue. We have now modified the illustration.

Since response phase can be defined in different ways, we no longer illustrate the response phase in Figure 1. Instead, we now show the time lag between neural responses in Figure 1D. Correspondingly, in the main analysis, we have now separately extracted the word and AM responses by time shifting and averaging the responses across σ1- and σ2-amplified conditions (please see Figure 4).

– While a time-warping analysis is being performed for data collected in Experiment 3, no mention of this approach is being explained in more details (as to possible interpretational limits) in the actual Results section.

Thank you for pointing out this issue. We have now detailed the time-warping procedure in Materials and methods.

“Time-Warping Analysis: In natural speech used in Experiment 3, syllables were not produced at a constant rate, and therefore the responses to syllables and words were not frequency tagged. […] The word index j ranged from 1 to the total number of syllables in a story.”

We have also mentioned the assumption underlying the time-warping procedure.

“In the time-warping procedure, it was assumed that the syllable response was time-locked to the syllable onsets and the word response was time-locked to word onsets. The frequency-domain analysis was subsequently applied to the time-warped response, following the same procedure as adopted in the analysis of the response to isochronous speech.”

In the time-domain analysis (please see Figure 6), which does not involve time warping, it is further confirmed that there is a word response synchronous to the word onset.

I am here particularly concerned about the phase analysis which is not being sufficiently reported in details to clearly understand the results and the claims of the authors.

The previous manuscript did not mention how the response phase was calculated. It was the phase of the complex-valued Fourier coefficient. Most of the phase results are now moved to Figure 3—figure supplement 1, and we have clearly defined the method for phase calculation. Since we analyzed the steady-state response, we did not calculate, e.g., the instantaneous phase.

In the Results, it is mentioned:

“The Fourier transform decomposes an arbitrary signal into sinusoids and each complex-valued Fourier coefficient captures the magnitude and phase of a sinusoid. […] Nevertheless, the response phase difference between the σ1- and σ2-amplified conditions carried important information about whether the neural response was synchronous to the word onsets or amplified syllables…”

In Materials and methods, it is mentioned:

“Then, the response averaged over trials was transformed into the frequency-domain using Discrete Fourier transform (DFT) without any additional smoothing window. […] The 2-Hz phase difference between the σ1- and σ2-amplified conditions was averaged over participants in each electrode.”

In line with this, I highlight two additional issues:– Subsection “Response phase at 2 Hz”, Figure 4: the authors argue that attention may explain the lack of phase differences. Additionally, the authors seem to assume in their working hypothesis that the phase response should be comparable across participants.In a recent study (Kösem et al., 2016), no evidence for phase-related linguistic parsing in low-frequency activity was found when participants actively or passively listened to a stationary acoustic speech sequence whose perceptual interpretation could alternate. Are these findings compatible with those illustrated in Figure 4? Furthermore, some important inter-individual differences were reported, which could severely impact the primary assumption of phase-consistency across individuals.

The reviewer raised two very important questions here. One was about the phase consistency across participants, and the other was about how the current results were related to the results in Kösem et al., 2016. In the following, we answered the two questions separately.

Inter-participant phase consistency

In terms of the experiment design, we do not assume that the response phase is consistent across individuals. We only hypothesize that there are two potential neural response components, which are separately synchronous to the speech envelope and the word rhythm. In the previous analysis, however, we did implicitly assume consistent response phase across individuals, since the response phase appeared to be consistent across individuals in most conditions (previous Figure 3). However, as the reviewer pointed out, this assumption was not necessary to test our hypothesis. Therefore, we added a new analysis that did not assume inter-participant phase consistency. This analysis calculated the phase difference between the σ1- and σ2-amplified conditions for individuals and tested whether the phase difference significantly deviated from 0º. This analysis was illustrated in Figure 3—figure supplement 1E.

Relation to the results in Kösem et al., 2016

We actually have our own pilot data acquired using a paradigm very similar to the paradigm adopted in Kösem et al., 2016. In the pilot study, a single word was repeated in the stimuli and we did not observe a word-related response either. It seems like a robust word synchronous response is only observable when the sequence presents different words instead of repeating the same word. There are several potential reasons for this difference. First, repeating the same word generates an acoustic rhythm of the same rate with the word rhythm. Neural tracking of the acoustic rhythm may potentially interact with the word response. Secondly, repetition of the same word may lead to neural adaptation and phenomena such as semantic satiation, which attenuates the neural response to words.

We have now added a discussion about the issue in *Discussion*.

“Furthermore, a recent study shows that low-frequency cortical activity cannot reflect the perception of an ambiguous syllable sequence, e.g., whether repetitions of a syllable is perceived as “flyflyfly” or “lifelifelife” (Kösem et al., 2016).”

“Although the current study and previous studies (Ding et al., 2018; Makov et al., 2017) observe a word-rate neural response, the study conducted by Kösem et al., 2016, does not report observable neural activity synchronous to perceived word rhythm. […] Therefore, it is possible that low-frequency word-rate neural response more strongly reflects neural processing of novel words, instead of the perception of a steady rhythm (see also Ostarek et al., 2020).”

– Have the authors considered the possibility that the choice of EEG reference may largely affect the reported mean instantaneous phase responses and if so, could they control for it?

We agree that the choice of EEG reference can influence the absolute phase. However, the hypothesis we would like to test is about the phase difference across conditions, which should not be sensitive to the choice of EEG reference.

To confirm that the phase difference between conditions is not strongly influenced by the choice of EEG electrodes, we have also analyzed the results using the average of sixty-four electrodes as the reference. The phase difference results are illustrated in Author response image 3, which is consistent with the results obtained using the average of the two mastoid electrodes as the reference (Figure 3 —figure supplement 1E).

**Author response image 3. sa2fig3:** Topographical distribution of the 2-Hz phase difference between the σ1- and σ2-amplified conditions using the average of sixty-four electrodes as the reference. The phase difference is calculated for each participant and each electrode, and then averaged over participants. The black dots indicate the electrodes showing a significant phase difference between the σ1- and σ2-amplified conditions (P < 0.05, bootstrap, FDR corrected).

Reviewer #2:[…] I find the framing and discussion a bit gappy and I missed multiple lines of related research. For instance, the literature on the relationship between delta-band oscillations and prosody is not mentioned in the Introduction; likewise, the literature on temporal prediction versus linguistic prediction is not discussed well. In addition, I also think that this study is not as definitive as the authors claim, resulting from the choice of unnatural (i.e., isochronous) stimuli that, strangely enough, the authors claim to be natural.

Thank you for pointing out these issues. We have now made substantial modifications to the Introduction and Discussion sections, and have modified the conclusions based on new data and new analyses. Please see our responses in the following for details.

Below, I given some comments for the authors to further improve their interesting work.1) False claim of studying natural speech: The authors artificially induce unnatural stimulus rhythms as a means to frequency-tag their stimuli and perform frequency-domain analysis. While this is an interesting approach, this results in stimuli that do not have ecological validity. Using unnatural stimuli with artificial periodicity to propose periodic electrophysiological processing of words is not scientifically sound. In natural speech, words do not occur periodically. Why assume then that they are processed by a periodic oscillatory mechanism? While the frequency-tagging approach inherently provides good SNR, it cannot claim ecological validity for anything above phonemes or syllables, as for these, there are no linguistic corpus studies out there that show periodicity of the constructs to start with. Specific comment:

Thank you for raising these important issues. First, we want to clarify that we did use natural speech in the “natural speech condition”. We time warp the neural response instead of the speech sound. The confusion is caused by the previous Figure 1C, which shows time-warped speech for illustrative purposes. We have now modified the figure. The updated Figure 1 only displays the stimuli and Figure 2E illustrates time-warped neural response. We have also included samples of the stimulus as Supplementary file 3.

We fully agree that words are not of equal duration in natural speech. Syllables and phonemes are not of equal duration either. The motivation of including a natural speech condition is to test whether neural activity is synchronous to words when the word rhythm is not constant. Therefore, we never used the word “oscillation” in the manuscript. We used frequency-tagging as a high SNR analysis method to extract the word-related response, but did not assume that words were encoded by periodic neural oscillations. We did refer to the word response as a “delta-band” response. However, we used the term to distinguish it from the “theta-band” syllabic-level response. In the current manuscript, we have avoided using the word “delta-band” to denote the word response.

Subsection “Neural tracking of words in natural spoken narratives”: "natural" The authors artificially make natural speech unnatural by making it isochronous. It is bizarre to still call the resulting stimulus "natural". Isochronous word sequences are certainly not natural speech. Hence, the authors can also not claim that they demonstrate cortical tracking of natural speech. To show this, they would have to show that there is speech-brain synchronicity for the non-isochronous condition. Certainly, to do so, the authors cannot use their standard spectral-peak analysis, but would have to show coherence or phase-locking against a different baseline (e.g., scrambled, autocorrelation-preserving speech). I strongly encourage the authors to do this analysis instead of pretending that isochronous word sequences are natural. The more general problem with this is that it adds to literature claiming that oscillations track words in human speech processing, which cannot possibly be true in this simplistic version: words are simply too variable in duration and timing.

The “natural speech” used in the study was not time-warped, which was explained in the response to the previous question. Nevertheless, metrical stories had an implicit disyllabic word-level rhythm, which was unnatural. The speaker reading the stories, however, were unaware of the purpose of the study, and the listeners were not told that some stories had a metrical word rhythm. We now conducted a behavioral experiment to test if these stories sounded natural and whether listeners could easily hear the metrical word rhythm. The results were reported in Supplementary file 1 (see Materials and methods for details). In short, most participants did not detect any difference between the metrical and nonmetrical stories and perceived the speech materials as natural.

“Thirty-four participants (19-26 years old, mean age, 22.5 years; 17 females) took part in a behavioral test to assess the naturalness of the stimuli…”

“…The test was divided into 2 blocks. In block 1, the participants listened to a metrical story and a nonmetrical story read by a human speaker, which were presented in a pseudorandom order. The stories were randomly selected from the story set. Each story ranged from 53 to 66 second in duration. After listening to each story, the participants were asked to write a sentence to summarize the story and fill out a questionnaire…”

“…In block 1, the first question in the questionnaire asked whether the two types of stories, a metrical and a nonmetrical story, showed any noticeable difference regardless of their content. […] The reasons reported included (1) exaggerated intonation (N = 2); (2) the speed and intonation pattern seemed uniform (N = 2); (3) lack of emotion (N = 2); (4) the pitch went up at the end of each sentence (N = 1). In sum, most participants thought the stories were naturally read and only two participants (6%) commented on the uniformity of pace.”

2) Depth of framing and discussion: (a) Introduction is not specific enough to speech prosody versus the different types of results on linguistic information (e.g., syntactic structure / phrasing, syntactic and lexical entropy), including reference to prior work; also, relevant references that suggested delta-band oscillations to serve both bottom-up (i.e., prosody) and top-down (i.e., syntax, prediction) functionalities are not discussed in the Introduction; (b) interpretation of the results in terms of the different proposed functionalities of the delta-band lacks depth and differentiation.

Thank you for the useful suggestions, we have updated the Introduction and Discussion. Please see, e.g., the response to the following question.

Introduction: What is missing here is mentioning that delta-band oscillations have also been implied in prosody (e.g., Bourguignon et al., 2013). Relatedly, the idea that delta-band oscillations are related both to slow-fluctuating linguistic representations (e.g., phrases; Meyer and Gumbert, 2018; Meyer et al., 2016; Weissbart et al., 2019) and prosody has been put forward various times (e.g., Ghitza, 2020; Meyer et al., 2016). I think that mentioning this would help the framing of the current manuscript: (1) delta = symbolic linguistic units, (2) delta = prosody, (3) current study: Is delta reflecting symbolic processing or prosody?

The reviewer made a very good point here and we have added the following into Introduction.

“Speech comprehension, however, requires more than syllabic-level processing. […] Therefore, it remains unclear whether cortical activity can synchronize to linguistic units in natural spoken narratives, and how it is influenced by bottom-up acoustic cues and top-down linguistic knowledge”.

Furthermore, we have added the following into the Discussion.

“It remains elusive what kind of mental representations are reflected by cortical responses synchronous to linguistic units. [...] Previous psycholinguistic studies have already demonstrated that syntax has a significant impact on prosody perception (Buxó-Lugo and Watson, 2016; Garrett et al., 1966).”

“In speech processing, multiple factors contribute to the word response and these factors interact. […] This result is consistent with the idea that prosodic cues have a facilitative effect on speech comprehension (Frazier et al., 2006; Ghitza, 2017, 2020; Giraud and Poeppel, 2012).”

Subsection “Time course of ERP response to words”: As to prediction and the delta-band, please consider discussing Weissbart et al., 2019, Meyer and Gumbert, 2018, Breska and Deouell, 2017, Lakatos et al., 2013, Roehm et al., 2009; in particular the relationship between linguistic predictions and the delta-band should be discussed shortly here. In general, there is need in the field to characterize the relationship between temporal prediction, prosody, linguistic units, and the delta-band. Just noting all this would be fine here.

Thank you for the useful suggestions and we have added the following to Discussion.

“It is also possible that neural activity synchronous to linguistic units reflect more general cognitive processes that are engaged during linguistic processing. […] In addition, low-frequency neural activity has also been suggested to reflect the perception of high-level rhythms (Nozaradan et al., 2011) and general sequence chunking (Jin et al., 2020).”

"semantic processing" Why semantics? See my above comment on the relationship between the delta-band and temporal predictions (e.g., Roehm et al., 2009; Stefanics et al., 2010). In addition, please consider the recent reconceptualisation of the N400-P600 complex in terms of predictive coding (e.g., Kuperberg et al., 2019).

We have replaced the sentence about “semantic processing” with the following sentence:

“This component is consistent with the latency of the N400 response, which can be observed when listening to either individual words or continuous speech (Broderick et al., 2018; Kutas and Federmeier, 2011; Kutas and Hillyard, 1980; Pylkkänen and Marantz, 2003; Pylkkänen et al., 2002).”

We have also added the following into the Discussion.

“The 400-ms latency response observed in the current study is consistent with the hypothesis that the N400 is related to lexical processing (Friederici, 2002; Kutas and Federmeier, 2011). Besides, it is also possible that the second syllable in a disyllabic word elicits weaker N400 since it is more predictable than the first syllable (Kuperberg et al., 2020; Lau et al., 2008).”

3) EEG preprocessing / report thereof does not accord to standards in the field. Apparently, the EEG data have not been cleaned from artifacts except for calculating the residual between the EEG and the EOG. Hence, I cannot assume that the data are clean enough to allow for statistical analysis. Specific point:Materials and methods: The EEG-cleaning section is very sparse. With standard EEG and standard cleaning, rejection rates for adult EEG are in the other of 25-35 %. It am highly doubtful that these EEG data have been cleaned in an appropriate way. There is no mentioning of artifact removal, amount of removed data segments, time- and frequency-domain aspects that were considered for artifact definition. What toolboxes were used, what algorithms?

Thank you for pointing out this issue. We have now mentioned the software and added more details about the pre-processing procedures. We only used basic Matlab functions without any particular toolbox.

“All preprocessing and analysis in this study were performed using Matlab

(The MathWorks, Natick, MA). […] Occasional large artifacts in EEG/EOG, i.e., samples with magnitude > 1 mV, were removed from the analysis (Jin et al., 2018).”

We have now removed the trials with obvious muscle artifacts after the preprocessing.

“To further remove potential artifacts, the EEG responses were divided into 7-s trials (a total of 148 trials for Experiments 1-3, and 104 trials for Experiment 4), and we visually inspected all the trials and removed trials with identifiable artifacts. On average, 8.45% ± 3.20% trials were rejected in Experiment 1, 15.20% ± 3.97% trials were rejected in Experiment 2, 10.35% ± 1.53% trials were rejected in Experiment 3, and 12.9% ± 4.46% trials were rejected in Experiment 4.”

4) Terminology: Like many people in the field, the authors speak of “tracking” of linguistic information. To be clear: While it is fine that the field understands the problems of the term “entrainment”, now opting for “tracking”, it does not make much sense to use the combination of “tracking” and “linguistic units”. This is because linguistic units are not in the stimulus; instead they are only present in the brain as such. They cannot be tracked; instead, they need to be “associated”, “inferred”, or even “generated”. If one were to assume that the brain “tracks” linguistic units, this would mean that the brain tracks itself. Specific comments:"cortical activity tracking of higher-level linguistic units" Linguistic units, such as "words and phrases" cannot be tracked, as they are not present in the stimulus. They are purely symbolic and they must be inferred. They cannot be sensed. Instead, they are present only in the mind and electrophysiology of the listener. Linguistic units are activated in memory or generated in some form. They are internal representations or states. If one were to say that the brain tracks these in some way, one would say that the brain tracks its own internal representations / states. Try and rephrase, please.

Thank you for pointing out this important terminology issue. We certainly agree that the word-related responses are internally constructed. The word “tracking” was used loosely in the previous manuscript. Now, we avoided calling the response a “word-tracking response”, and referred to the response as a word response or a neural response synchronous to the word rhythm.

"linguistic information" cannot be extracted from acoustic features. Instead, linguistic information (e.g., distinctive features in phonology, semantic meaning, part-of-speech labels) must be associated with a given acoustic stimulus by an associative / inferential cognitive process. When we hear some acoustic stimulus, we associate with something linguistic. Just as in my above comment on the loose use of the term “tracking”, I suggest the authors to change this wording as well. A suggestion for a better phrasing would be "Linguistic information is retrieved / constructed / generated when specific acoustic information is encountered / recognized…". Or "Speech comprehension involves the active inference of linguistic structure from speech acoustics".

We have removed the description, and we no longer use the word “extract” in similar situations.

Discussion: "tracks" see my above comment.

Please see our reply to the previous question.

5) Potential differences between the σ1 and 2 conditions: There are apparent differences in the number of sensors that show sensitivity to the amplitude modulation. This is not captured in the current statistical analysis. Still, given these drastic differences, the claim that acoustics of speech prosody do not affect delta-band amplitude is certainly much too strong and may just result from a lack of analysis (i.e., overall phase / ERP rather than sensor-by-sensor analysis). See specific comment next:

We have updated our conclusions based on a new analysis. Please see our reply to the editorial comments.

"or the phase of the 2-Hz amplitude modulation" I disagree with excitement and a happy suggestion that the authors must consider. Look, the plots (Figure 3, σ1/2) may not show differences in the phase angle. Yet, what they may point to is a dimension of the data that the authors have not assessed yet: topographical spread. From simply eyeballing it is clear that the number of significant electrodes differs drastically between the σ1 and σ2 conditions. This pattern is interesting and should be examined more closely. Is word-initial stress more natural in Chinese? If so, this could be interpreted in terms of some influence of native prosodic structure: Amplitude modulation does appear affect the amount / breadth of EEG signal (i.e. number of sensors), but only if it conforms to the native stress pattern? Not only words / phrases are part of a speakers linguistic knowledge, but also the stress pattern of their language. I urge the authors to look into this more closely.

The reviewer raised a very important point about the stress pattern in Chinese. According to the classic study by Yuen Ren Chao (Mandarin Primer, Harvard University Press, 1948), the stress of disyllabic Chinese words usually falls on the second syllable. Later studies more or less confirm that slightly more than 50% of the disyllabic words tend to be stressed on the second syllable. However, it is well established that, in Chinese, stress is highly dependent on the context and does not affect word recognition. For example, Shen, 1993, noted that “However, unlike English where lexical stress is fixed and can be predicted by the phonology, lexical stress in Mandarin varies socio-linguistically and idiosyncratically. Some disyllabic words can be uttered iambically or trochaically”.

Since linguistic analysis does not generate a strong prediction about whether σ1-amplified or σ2-amplified speech should sound more natural, we now have asked a group of participants (*N* = 34) to rate the naturalness of the speech designed for these two conditions (see Materials and methods for details). The results showed that half of the participants commented that speech in the two conditions were equally natural (*N* = 17). For the other half of the participants, most thought σ1-amplified speech sounded more natural (*N* = 15) while the others thought σ2-amplified speech was more natural (*N* = 2).

In sum, while previous linguistic studies tend to suggest that σ2-amplified speech is slightly more natural, our behavioral assessment suggests that σ1-amplified speech is slightly more natural. Therefore, it is difficult to draw a solid conclusion about which condition is more consistent with natural speech.

Additionally, the response power does not differ between the σ1-amplified or σ2-amplified conditions, and only the inter-participant phase coherence differ between conditions, as mentioned by the reviewer. This phenomenon, however, can potentially be explained without making any assumption about the interaction between the AM and word responses: If the AM and word responses are independent, the measured neural response is the sum of the two responses and its phase is influenced by both components. If the phase of the AM response is more consistent with the phase of the word response in the σ1-amplified condition, the inter-participant phase coherence will be higher in the σ1-amplified condition on the assumption that the strength of the AM and word response varies across participants (illustrated in Author response image 4).

**Author response image 4. sa2fig4:** Illustration of the response phase for individuals. The red and blue arrows indicate the phase of word response and AM response, which are assumed to be consistent across individuals. The AM response is 180° out of phase between the σ1 and σ2-amplified conditions, while the word response phase is the same in both conditions. The measured response is the vector sum of the AM and word responses. The purple arrows indicate the phase of the measured response for individual participants. If the phase of the AM response is more consistent with the phase of the word response in the σ1-amplified condition, the inter-participants phase coherence is higher for the σ1-amplified condition than the σ2-amplified condition.

Therefore, in the current study, we integrate the σ1- and σ2-amplified conditions to separate the AM and word responses, and future studies are needed to establish whether the neural response is modulated by the naturalness of speech prosodic cues. We have now added discussions about this issue.

“Finally, it should be mentioned that we employed amplitude modulation to manipulate the speech envelope, given that the speech envelope is one of the strongest cues to drive stimulus-synchronous cortical response. […] Future studies are needed to characterize the modulation of language processing by different prosodic cues and investigate the modulatory effect across different languages.”

"The response" see my above comment on the differences in the number significant sensors in the σ1 and 2 conditions.

The sentence is replaced as the following sentence:

“Consistent with previous findings, in the current study, the 4-Hz syllable response was also enhanced by cross-modal attention (Figure 3B). The 2-Hz AM response power, however, was not significantly modulated by cross-modal attention (Figure 4D, and Figure 5B), suggesting that attention did not uniformly enhance the processing of all features within the same speech stream…”

Reviewer #3:[…] I enjoyed reading this manuscript, which tackles an interesting and topical question. And I would like to commend the authors on a very nice set of experiments – it's a lovely design.However, I also had a number of fairly substantial concerns – primarily about the conclusions being drawn from the data.1) My main concern with the manuscript is that it seems – a priori – very committed to discussing the results in terms of tracking by activity in a specific (i.e., delta) cortical band. I understand the reason to want to do this – there is a lot of literature on speech and language processing that argues for the central role of cortical oscillations in tracking units of speech/language with different timescales. However, in the present work, I think this framework is causing confusion. To argue that delta tracking is a unitary thing that might wax and wane like some kind of push/pull mechanism to track acoustics sometimes and linguistic features other times seems unlikely to be true and likely to cause confusion in the interpretation of a nice set of results. My personal bias here – and I think it could be added for balance to the manuscript – is that speech processing is carried out by a hierarchically organized, interconnected network with earlier stages/lower levels operating on acoustic features and later stages/higher levels operating on linguistic features (with lots of interaction between the stages/levels). In that framing, the data here would be interpreted as containing separable simultaneous evoked contributions from processing at different hierarchical levels that are time locked to the relevant features in the stimulus. Of course, because of the nature of the experiments here, these contributions are all coming at 2 Hz. But to then jump to saying a single "delta oscillation" is preferentially tracking different features of the speech given different tasks, seems to me to be very unlikely. Indeed, the authors seem sensitive to the idea of different evoked contributions as they include some discussion of the N400 later in their manuscript. But then they still hew to the idea that what they are seeing is a shift in what "delta-band" activity is doing. I am afraid I don't buy it. I think if you want to make this case – it would only be fair to discuss the alternatives and more clearly argue why you think this is the best way to interpret the data.

The reviewer raised a very important point about how to interpret the results. We actually agree with the reviewer’s interpretation. It might be our writing that has caused confusions. We do not think there is a unitary delta oscillation and the term “oscillation” is never used in the manuscript. In the current manuscript, we no longer refer to the results as “delta-band responses”, and we have revised the Introduction and Discussion sections to make it clear that responses from multiple processing stages can contribute to the measured EEG response.

Furthermore, a new analysis now separates the measured response into a word response and an AM response , which is also based on the assumption that multiple response components coexist in the measured response. Please see our reply to the editorial comments for details.

2) Following my previous point, I have to admit that I am struggling to figure out how, precisely, you are determining that "cortical activity primarily tracks the word rhythm during speech comprehension". There is no question that the cortical activity is tracking the word rhythm. But how are you arguing that it is primarily word rhythm. I am guessing it has something to do with Figure 3? But I am afraid I am not convinced of some of the claims you are making in Figure 3. You seem to want to embrace the idea that there is no difference in phase between the σ1-amplified and σ2-amplified conditions and that, therefore, the signal is primarily tracking the word rhythm. But I think there is an important flaw in this analysis – that stems from my concern in point number 1 above. In particular, I think it is pretty clear that there are differences in phase at some more frontal channel locations, and no difference in more posterior channels (Figure 3A, rightmost panel). So, going back to my issue in point number 1, I think it is very likely that word rhythm is being tracked posteriorly (maybe driven by N400 like evoked activity) – and that phase is similar (obviously). But it also seems likely that the 2 Hz rhythm over frontal channels (which likely reflect evoked activity based on acoustic features) are at a different phase. Now, cleanly disentangling these things is difficult because they are both at 2Hz and will contaminate each other. And I think when subjects are watching the silent video, the posterior (N400 like) contribution disappears and all you have left is the acoustic one. So, again, I think two separable things are going on here – not just one delta rhythm that selectively tracks different features.

Thank you for the insightful comments and please see our reply to the editorial comments.

3) I'm afraid I must take issue with the statements that "Nevertheless, to dissociate linguistic units with the related acoustic cues, the studies showing linguistic tracking responses mostly employ well-controlled synthesized speech that is presented as an isochronous sequence of syllables… Therefore, it remains unclear whether neural activity can track linguistic units in natural speech, which is semantically coherent but not periodic, containing both acoustic and linguistic information in the delta-band." I know the (very nice) studies you are talking about. But you have just also cited several studies that try to disentangle linguistic and acoustic features using more natural speech (e.g., Brodbeck; Broderick). So I think your statement is just too strong.

Thank you for pointing out this issue. We have now revised the Introduction to make it more precise.

“Previous studies suggest that low-frequency cortical activity can also reflect neural processing of higher-level linguistic units, e.g., words and phrases (Buiatti et al., 2009; Ding et al., 2016a; Keitel et al., 2018), and the prosodic cues related to these linguistic units, e.g., delta-band speech envelope and pitch contour (Bourguignon et al., 2013; Li and Yang, 2009; Steinhauer et al., 1999). […] It remains to be investigated, however, how bottom-up prosodic cues and top-down linguistic knowledge separately contribute to the generation of these word-related responses”.

4) I was confused with the presentation of the data in Figure 2F and G. Why are some bars plotted up and others down? They are all positive measures of power, and they would be much easier to compare if they were all pointed in the same direction. Unless I am missing the value of plotting them this way?

In the previous manuscript, we thought it was easier to make comparisons if the bars were presented as mirror images. However, it was indeed confusing to have downward bars. We have now made all bars pointed in the same direction (Figure 3A, B).